# An inorganic-blended p-type semiconductor with robust electrical and mechanical properties

You Meng [1,2,8], Weijun Wang[1,8], Rong Fan[3,4,8], Zhengxun Lai[1], Wei Wang[1], Dengji Li[1], Xiaocui Li [3], Quan Quan[1], Pengshan Xie[1], Dong Chen[1], He Shao[1], Bowen Li[1], Zenghui Wu[1], Zhe Yang[5], SenPo Yip[6], Chun-Yuen Wong [5] ✉, Yang Lu [4,7] ✉ & Johnny C. Ho [1,2,6] ✉

Inorganic semiconductors typically have limited p-type behavior due to the scarcity of holes and the localized valence band maximum, hindering the progress of complementary devices and circuits. In this work, we propose an inorganic blending strategy to activate the hole-transporting character in an inorganic semiconductor compound, namely tellurium-selenium-oxygen (TeSeO). By rationally combining intrinsic p-type semimetal, semiconductor, and wide-bandgap semiconductor into a single compound, the TeSeO system displays tunable bandgaps ranging from 0.7 to 2.2 eV. Wafer-scale ultrathin TeSeO films, which can be deposited at room temperature, display high hole field-effect mobility of 48.5 cm²/(Vs) and robust hole transport properties, facilitated by Te-Te (Se) portions and O-Te-O portions, respectively. The nanosphere lithography process is employed to create nanopatterned honeycomb TeSeO broadband photodetectors, demonstrating a high responsibility of 603 A/W, an ultrafast response of 5 μs, and superior mechanical flexibility. The p-type TeSeO system is highly adaptable, scalable, and reliable, which can address emerging technological needs that current semiconductor solutions may not fulfill.

Semiconducting properties are present in numerous elements and compounds. Some examples of pure elements are found in group 14, such as the commercially important C[1], Si[2], and Ge[3], which are connected by covalent bonds. By sharing their outermost (valence) electrons to form a full electron shell, the stable balance of attractive/repulsive forces among atoms allows them to attain stable electronic configuration. Nonetheless, covalent bonding relies heavily on the directional nature of atomic orbitals, imposing substantial constraints

on the design of covalent compounds. For instance, it only takes 4% lattice constant mismatch to make the Si-Ge system strained and possibly metastable[4,5]. Moreover, owing to their stable crystal structures and chemically unreactive properties, producing high-quality covalent semiconducting materials invariably requires high-temperature growth (600 °C or higher) and subsequent annealing procedures[6,7]. These factors limit the feasibility of tuning the bandgap and the flexible integration of covalent semiconductors.

[1]Department of Materials Science and Engineering, City University of Hong Kong, Kowloon 999077, Hong Kong SAR. [2]State Key Laboratory of Terahertz and Millimeter Waves, City University of Hong Kong, Kowloon 999077, Hong Kong SAR. [3]Department of Mechanical Engineering, City University of Hong Kong, Kowloon 999077, Hong Kong SAR. [4]Chengdu Research Institute, City University of Hong Kong, Chengdu 610200, China. [5]Department of Chemistry, City University of Hong Kong, Kowloon 999077, Hong Kong SAR. [6]Institute for Materials Chemistry and Engineering, Kyushu University, Fukuoka 816 8580, Japan. [7]Department of Mechanical Engineering, The University of Hong Kong, Pokfulam 999077, Hong Kong SAR. [8]These authors contributed equally: You Meng, Weijun Wang, Rong Fan. ✉e-mail: acywong@cityu.edu.hk; ylu1@hku.hk; johnnyho@cityu.edu.hk

On the other hand, with sharply different electronegativities, the electrostatic attraction between oppositely charged ions could form ionic or polar covalent bonds. Compound semiconductors made by these two types of chemical bonds, primarily comprising transition-metal cations from group 10–14, have been identified to possess respectable electron mobility, i.e., tens of cm²/(Vs) for amorphous/polycrystalline and hundreds of cm²/(Vs) for crystalline ones[8–10]. However, despite extensive efforts over many years, the inferior hole mobility exhibited by their p-type counterparts has hindered the progress of complementary devices and circuits. The main causes for their poor p-type performance include the localized valence band maximum (VBM), strong self-compensation effect, and poor material stability[8,11,12]. For instance, in p-type transition-metal oxides (halides) such as $CuO_x$, SnO, and CuI, high concentrations of oxygen (halogen) vacancies typically function as compensating intrinsic defects that capture holes, thereby impeding their hole transport[13,14]. To bypass these ingrained issues, an alternative material design strategy is expected to explore better p-type semiconductors.

In this work, a high-mobility and air-stable p-type semiconducting system, namely tellurium-selenium-oxygen (TeSeO), was explored based on an inorganic blending strategy (Fig. 1a). Group 16 elements, i.e., Te, Se, and O, are utilized because of their similar $s^2p^4$ electronic

configurations and atomic radii, which endows good miscibility to form the TeSeO system. Due to the different ionization energy and electronegativity, the metallic characters of elements become more pronounced as one moves down group 16, which spans from insulator to semiconductor to metalloid. This variation allows for continuous tuning of the physical properties of the TeSeO system through different compositions. In particular, semimetal Te has high intrinsic hole mobility (~$10^5$ cm²/Vs) and small effective hole mass (~0.3 $m_0$)[15,16], which is covalently bonded as Te–Te or Te–Se that mainly contributes to the good hole-transport property. Semiconducting Se is used to effectively modify the bandgap (from 0.7 to 2.2 eV), oxidative ability, and crystallinity of TeSeO to meet technical requirements. Notably, the employment of O forms O–Te–O (polar covalent bonds) with a wider bandgap[17,18], endowing superior operating durability to TeSeO that is unachievable by other p-type thin-film semiconductors.

## Results

### TeSeO synthesis and microstructure characterization

To satisfy the scalable processing, in this work, TeSeO thin films were produced by room-temperature physical vapor deposition (PVD) combined with post-oxygen implantation (details shown in the Methods section). The crystal structures of TeSeO thin films were

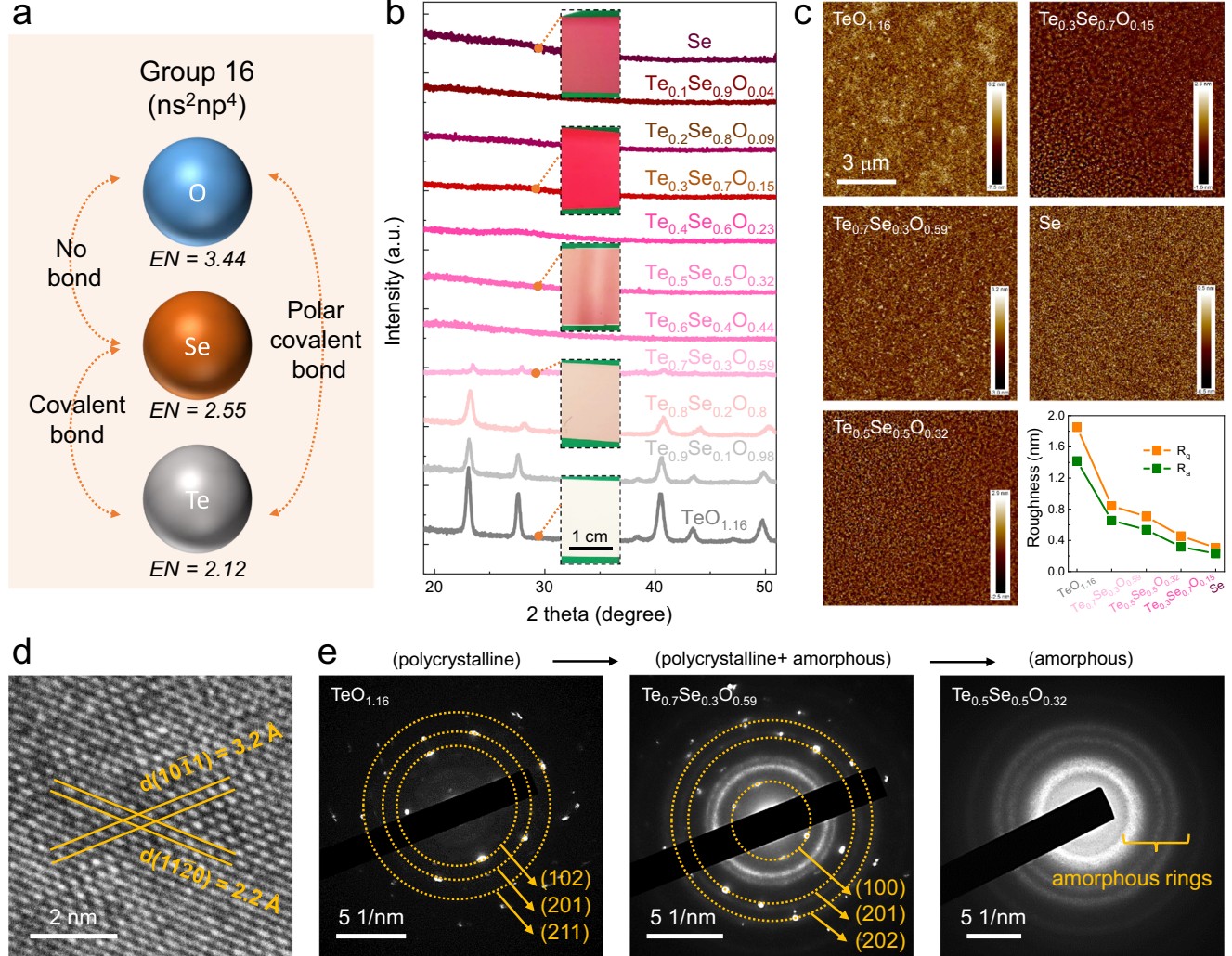

Fig. 1 | TeSeO synthesis and microstructure characterization. a Schematic of p-type TeSeO semiconducting system containing group 16 elements. b GIXRD patterns and c AFM images of TeSeO thin films with different compositions. Insets of b display the corresponding sample photographs. Inset of c shows the corresponding roughness values of TeSeO thin films based on root-mean-square deviation ($R_q$) and arithmetic-mean deviation ($R_a$). d Cross-sectional HRTEM image of $Te_{0.7}Se_{0.3}O_{0.59}$ thin film. e SAED patterns of $TeO_{1.16}$, $Te_{0.7}Se_{0.3}O_{0.59}$, and $Te_{0.5}Se_{0.5}O_{0.32}$ thin films.

characterized by a grazing-incidence x-ray diffractometer (GIXRD). As shown in Fig. 1b, the TeSeO samples with relatively high Te content (Te:Se ≥ 7:3) show a polycrystalline nature, while the Se-rich samples (Te:Se ≤ 6:4) are found to be amorphous. The diffraction peaks located at 23.1, 27.6, 40.5, and 49.7° agree well with those of (100), (101), (110), and (201) planes, respectively, of hexagonal system with P3$_1$21 [152] space group that is composed of chalcogen chains along the c axis[17]. All the diffraction peak positions slightly shift to higher angles with increasing Se content (Supplementary Fig. 1), indicating the decrease of the lattice constant. At the same time, the increased full width at half-maximum of the diffraction peaks also reveals the suppressed material crystallinity. As a strong glass former, the further addition of Se contents (Te:Se ≤ 6:4) completely disturbs the crystallization process of TeSeO, leading to the polycrystalline-to-amorphous phase transition. Remarkably, with the vigorous compositional changing of TeSeO thin films, the samples undergo significant color changes from metallic luster for TeO$_{1.16}$ to dark red for Se-rich samples, as the photographs depicted in the insets of Fig. 1b.

The Raman spectroscopy presented in Supplementary Fig. 2 identified three first-order Raman active Te/Se helical chain modes, including $E_1$ transverse (TO) phonon mode, $A_1$ mode, and $E_2$ mode[16]. Also, Se substituted for Te gives rise to an increase in the stretching frequency and the broadening of Raman bands. The undiscovered O-related Raman vibrations suggest the disordered nature of corresponding oxides that cannot efficiently enable the inelastic scattering of photons. To directly check the elemental distribution, a combination of energy-dispersive X-ray spectroscopy (EDS) mapping and X-ray photoelectron spectroscopy (XPS) were conducted on TeSeO thin films (Supplementary Figs. 3 and 4), where the uniform elemental distributions of Te, Se, and O are observed across the probed region. Moreover, the surface morphologies of TeSeO thin films with different composition ratios were examined by atomic force microscopy (AFM) with a scanning area of 10 × 10 μm (Fig. 1c). All the TeSeO thin films are smooth, uniform, and crack-free, which is crucial for practical devices. The extracted arithmetic-mean deviation of roughness decreases from 1.8 nm to 0.3 nm with increasing Se content (inset of Fig. 1c), mainly due to the decreasing grain sizes.

The microscopic structures of the TeSeO thin films were further analyzed by cross-sectional high-resolution transmission electron microscopy (HRTEM), as depicted in Fig. 1d. The HRTEM image shows clear lattice fringes with lattice spacings of 3.2 Å and 2.2 Å for Te$_{0.7}$Se$_{0.3}$O$_{0.59}$, which corresponds to the (10$\bar{1}$1) and (11$\bar{2}$0) crystalline planes of hexagonal Te/Se, respectively. The corresponding selected-area electron diffraction (SAED) pattern of Te$_{0.7}$Se$_{0.3}$O$_{0.59}$ shows a few diffraction spots (Fig. 1e), further indicating its polycrystalline structure. The observed characteristic diffuse halo, particularly evident in Te$_{0.5}$Se$_{0.5}$O$_{0.32}$, indicates that the addition of Se induces a polycrystalline-to-amorphous phase transition in TeSeO. Overall, no O-related phase diffraction pattern was detected in the SAED study, suggesting an amorphous state of oxides in TeSeO. The above microstructure analysis agrees well with that from XRD and Raman studies.

## Chemical bonding and band structure of TeSeO

To obtain the details of chemical bonding and atomic coordination in TeSeO thin films, core energy level spectra of Te 3$d$, O 1$s$, and Se 3$d$ were studied using XPS (Fig. 2a–c). All three XPS core energy levels of TeSeO exhibit redshifts of up to 800 meV with increasing Se content, resulting from the electron injection process. In detail, the Te$^{4+}$ and Te$^0$ peaks coexist in the Te 3$d$ spectra of TeSeO thin films (Fig. 2a), which means the partial oxidation of Te[18]. The corresponding Te$^{4+}$/(Te$^{0+}$+Te$^{4+}$) ratios decrease from ~58% (TeO$_{1.16}$) to ~25% (Te$_{0.3}$Se$_{0.7}$O$_{0.15}$) with increasing Se content (Supplementary Fig. 5 and Supplementary Table 1), revealing that the Se content could suppress the binding process between Te and O. At the same time, no Se$^{4+}$ (typically around

60 eV) peak is found from Se 3$d$ spectra in Fig. 2c, mainly owing to its larger electronegativity (2.55) than that of Te (2.1), which make it difficult to react with oxygen molecules to form SeO$_2$. The distinct peaks observed around 530.2 eV in the O 1$s$ spectra imply the O mainly acts as lattice oxygen species of O−Te−O (Fig. 2b and Supplementary Fig. 6). Generally, the adsorbed oxygen or hydroxyl group has higher binding energy around 532 eV[10], which is not witnessed in our TeSeO films. The gathered information on chemical bonding in inorganic-blended TeSeO, including Te-Te, Te-Se, and O-Te-O, is summarized in Table 1. Overall, the Se-regulating Te oxidation provides a window to change the material compositions among TeO$_2$ (p-type wide-bandgap semiconductor)[19,20], Te$_x$Se$_y$ (p-type semiconductor)[21,22], and Te (p-type semimetal)[16,17], and thus modify their corresponding physical/chemical properties in a wide range.

In general, continuously tuning bandgaps and band-edge energies in conventional p-type semiconductors is difficult[13]. Apart from the low formation energy of the electron donor, incorporating foreign atoms could inevitably perturb the host lattice thermodynamic equilibrium, possibly counteracting the p-doping effect[14]. These factors restrict the tuning feasibility on hole density and mobility of conventional p-type thin films. This work applies an inorganic blending strategy on the p-type TeSeO system, which combines intrinsic p-type semimetal, semiconductor, and wide-bandgap semiconductor in a single compound. As a result, the p-type TeSeO could be manufactured into a scalable thin-film form with reliable and tunable material properties. As presented in Fig. 2d, the optical bandgaps of TeSeO thin films were extracted from the absorption spectra by using the Tauc plot method. With increasing Se content, the energy bandgaps of TeSeO were broadening, with corresponding bandgap values monotonically shifting from 0.7 eV to 2.2 eV. The continuously tunable bandgaps achieved from TeSeO thin films cover ultraviolet (UV), visible, and short-wave infrared (SWIR) regions, revealing potentials in high-mobility p-channel transistors, solar cells, wideband photodetectors, etc.

The quantitatively predictable and available band structures of the TeSeO system are the prerequisite for device-level engineering. To this aim, the electronic structure variations of TeSeO thin films were identified by ultraviolet photoemission spectroscopy (UPS). The positions of Fermi energy levels and VBM levels could be extracted from the secondary electron cut-off region (Fig. 2e) and valence-band region (Fig. 2f), respectively[23]. Besides, the conduction band minimum (CBM) was calculated by subtracting the bandgap of each sample from its VBM. The corresponding energy band diagrams of TeSeO samples are presented in Fig. 2g. Obviously, for all TeSeO samples, their Fermi energy levels are always underlying the mid-point of bandgaps and relatively close to the valance band, which means that the inorganic-blended TeSeO system keeps p-type electrical properties. It is also found that the energy level of the VBM shifts faster with compositions than that of CBM. Fitting these results to the modified Vegard's law, the energy changes of VBM energies are nearly linear with the composition, while the CBM energies change with a strong bowing of ~1 eV. This result could be explained by the synergistic effect of Se substitution and partly oxidation in Te[22].

## Transport properties and electrical robustness of TeSeO

To investigate the hole-transport properties of TeSeO thin films, a series of bottom-gate top-contact (BGTC) thin-film transistors (TFTs) are constructed on SiO$_2$/p$^+$-Si wafers with Ni as source/drain electrodes. To guarantee low gate leakage currents and reliable parameter extraction, both channel layers and electrodes are patterned (inset in Fig. 3a). The channel thickness of ~10 nm, which is verified by cross-sectional STEM, is used to balance the conductivity and the on/off current ratio. As the corresponding electrical characteristics of Te$_{0.7}$Se$_{0.3}$O$_{0.59}$, Te$_{0.8}$Se$_{0.2}$O$_{0.8}$, and Te$_{0.9}$Se$_{0.1}$O$_{0.98}$ are shown in Fig. 3, typical p-channel transistor behaviors were observed for those devices, agreeing well with their energy band structures. The Se-poor

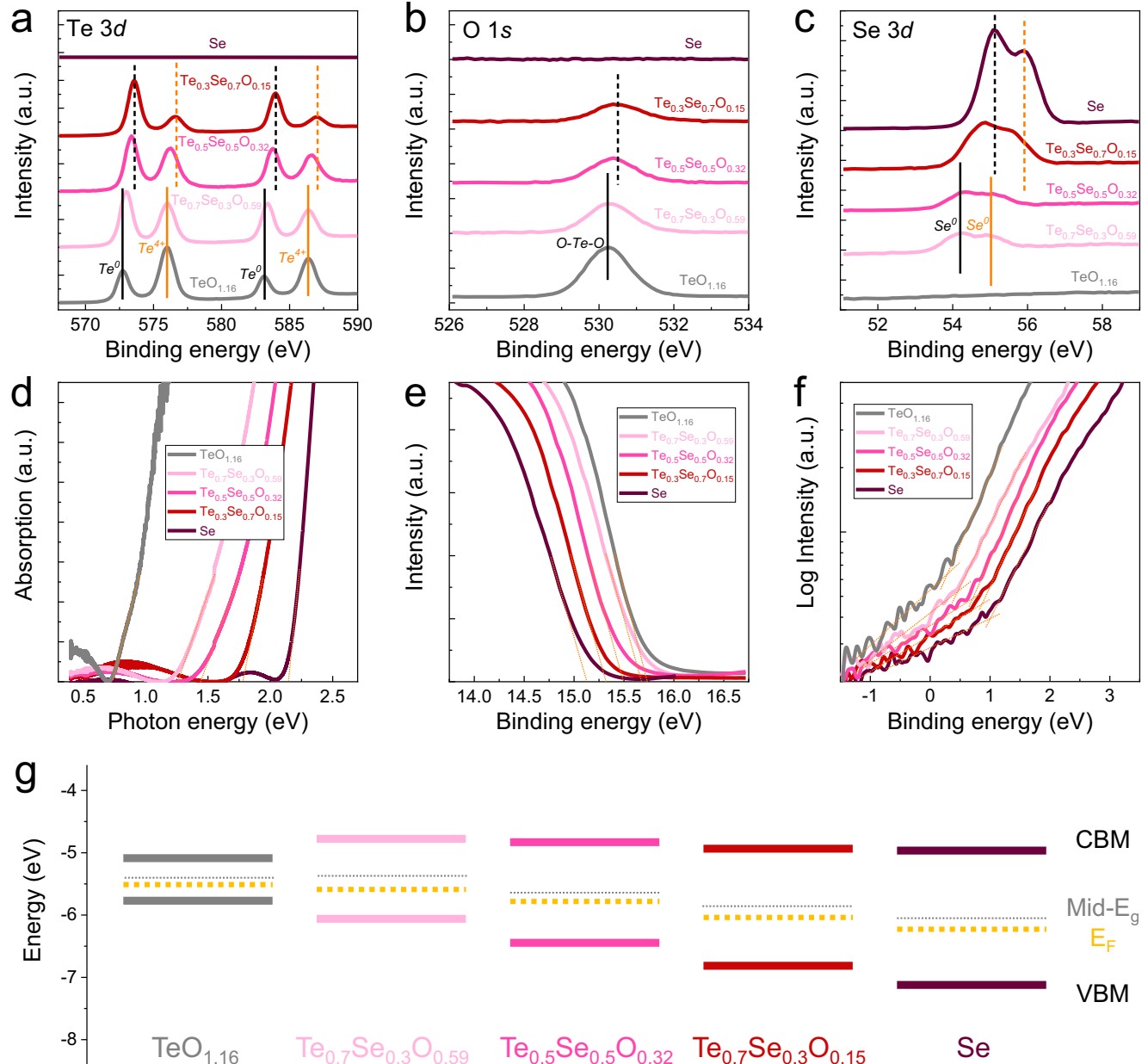

**Fig. 2 | Chemical bonding and band structure of TeSeO.** XPS (**a**) Te 3*d*, **b** O 1*s*, and **c** Se 3*d* analysis of TeSeO thin films with different compositions. **d** Absorption spectra of the prepared films with different compositions. UPS spectra of TeSeO thin films at **e** secondary electron cut-off regions and **f** valence-band regions. **g** Energy band diagram of the TeSeO thin films with changing compositions.

**Table 1 | The chemical bonding information of inorganic-blended TeSeO**

| Bond | Type | Bandgap | Character | Function |
|------|------|---------|-----------|----------|
| Te–Te | Covalent | 0.31 eV | p-type semimetal | High hole mobility |
| Te–Se | Covalent | 0.31~1.87 eV | p-type semiconductor | Bandgap modulation |
| O–Te–O | Polar covalent | 3.7 eV | p-type wide-bandgap semiconductor | Stability enhancement |

$Te_{0.9}Se_{0.1}O_{0.98}$ TFT exhibited high conductance and "always-on" transistor operation, which reflected high hole concentration in the channel layer that could not be completely depleted[23]. The Se alloying reduced the current level and mobility and shifted threshold voltage ($V_{TH}$) in the negative direction (Fig. 3d–f). Meanwhile, the Hall effect measurements of TeSeO films show a similar trend to the electrical properties observed in the TFTs study (Supplementary Table 2).

Notably, among all the samples, the $Te_{0.8}Se_{0.2}O_{0.8}$ TFT showed well-optimized electrical performance (Fig. 3e), including a high hole field-effect mobility ($\mu_{FE}$) of 48.5 cm²/(Vs) while maintaining a high $I_{on}/I_{off}$ ratio of ~10⁵. As summarized in Table 2 and Supplementary Table 3, such $\mu_{FE}$ and $I_{on}/I_{off}$ surpass most previously reported conventional scalable p-channel TFTs, including metal oxides, metal halides, perovskite halides, and organic materials[11]. Besides, the negligible counterclockwise hysteresis indicates the small amounts of electrical traps

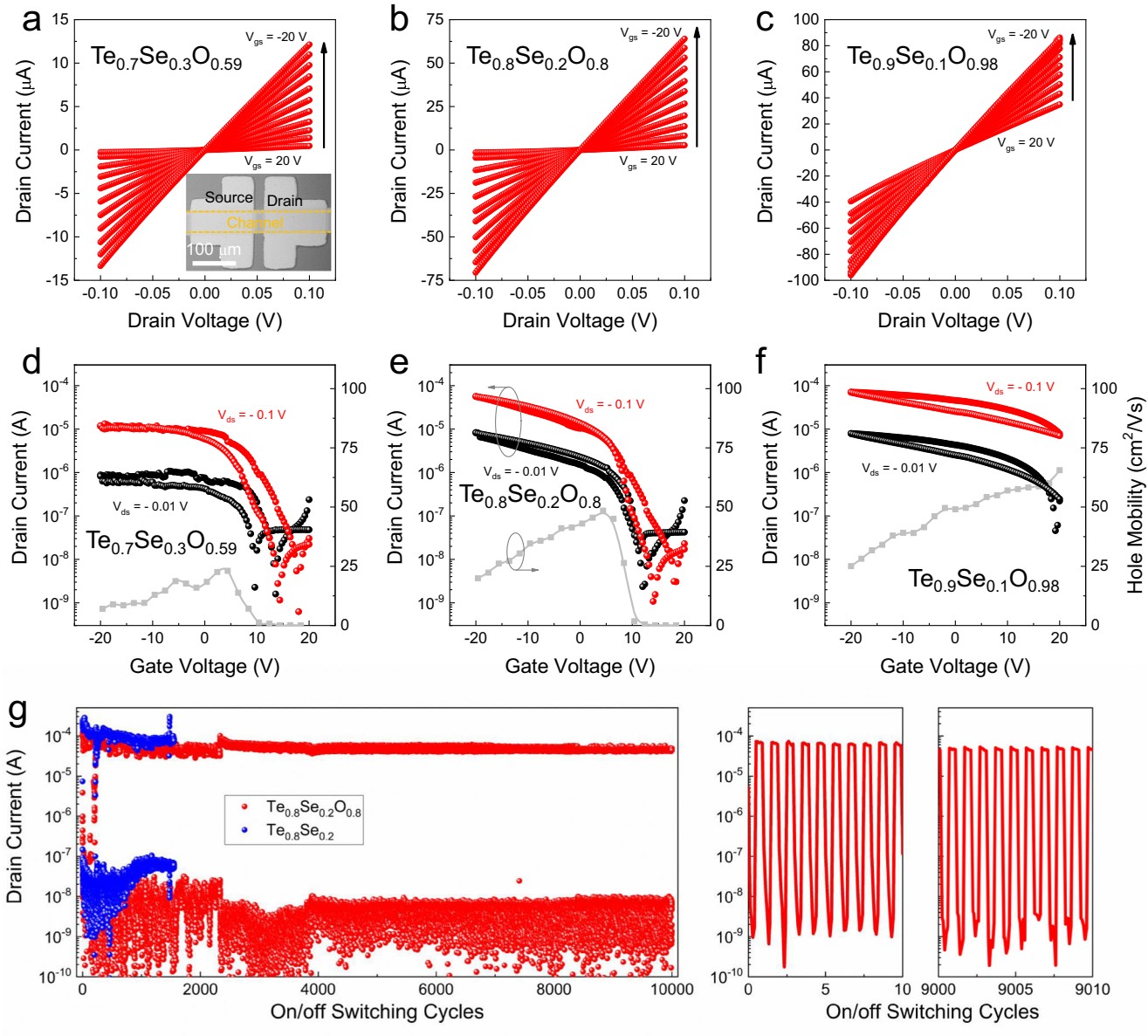

**Fig. 3 | Transport properties and electrical robustness of TeSeO.** Output curves of **a** $Te_{0.7}Se_{0.3}O_{0.59}$, **b** $Te_{0.8}Se_{0.2}O_{0.8}$, and **c** $Te_{0.9}Se_{0.1}O_{0.98}$ TFTs. Inset of **a** shows the optical image of the device. Transfer curves and hole mobilities of **d** $Te_{0.7}Se_{0.3}O_{0.59}$, **e** $Te_{0.8}Se_{0.2}O_{0.8}$, and **f** $Te_{0.9}Se_{0.1}O_{0.98}$ TFTs. **g** On/off switching measurements of $Te_{0.8}Se_{0.2}O_x$ and $Te_{0.8}Se_{0.2}$ TFTs. Inset of **g** shows the representative on/off switching cycles.

within TeSeO or at the interface between the channel and dielectric layers[24]. To study the scalability and uniformity of TeSeO TFTs, wafer-scale TFT arrays (10 × 10 array) are fabricated, and their statistical distribution of device performance is displayed in Supplementary Fig. 7. The TFT array shows 100% device yield with a hole mobility of 48.2 ± 8.4 cm²/(Vs), $I_{on}/I_{off}$ of $10^4$-$10^5$, and $V_{TH}$ of −4.2 ± 1.3 V. Such highly uniform electrical performance on the wafer scale is of high significance in the scalable applications of p-type thin-film semiconductors.

After successfully investigating the intrinsic transport properties of TeSeO thin films, the operational stability in ambient was also checked. As shown in Fig. 3g, with 10,000 times on/off switching, the $Te_{0.8}Se_{0.2}O_{0.8}$ device maintains its output current and good current modulation ability. Meanwhile, a control experiment was also carried out on $Te_{0.8}Se_{0.2}$ samples without oxygen implantation. After 1500 times on/off switching, the $Te_{0.8}Se_{0.2}$ device loses its transistor performance, possibly because of the electrical/thermal induced phase segregation[21,25]. Negative-bias stress (NBS) testing was also investigated on TeSeO thin films (Supplementary Fig. 8). After being gated at −20 V for 2 h, the corresponding $V_{TH}$ shifted negatively from −5 to −8.8 V without noticeable subthreshold swing variation under NBS. Using the equation of $\sigma_N = C_i \Delta V_{GS}/2e$[26], the charge-trapping states density ($\sigma_N$) was calculated to be -8.2 × 10¹¹ cm⁻², indicating the defect-state creation is negligible within the NBS test. In addition, benefiting from the partial oxidation in TeSeO thin films that could block the environmental influences, the devices exhibited stable operational

## Table 2 | Performance parameters of TeSeO p-channel TFTs

| Materials | $I_{on}$ [µA] | $I_{on}/I_{off}$ | $\mu_{FE}$ [cm²/(Vs)] | $V_{TH}$ [V] |
|---|---|---|---|---|
| $Te_{0.9}Se_{0.1}O_{0.98}$ | 72 | ~$10^2$ | 65.5 | 25 |
| $Te_{0.8}Se_{0.2}O_{0.8}$ | 56 | ~$10^5$ | 48.5 | 7.9 |
| $Te_{0.7}Se_{0.3}O_{0.59}$ | 18 | ~$10^4$ | 23.1 | 5.3 |

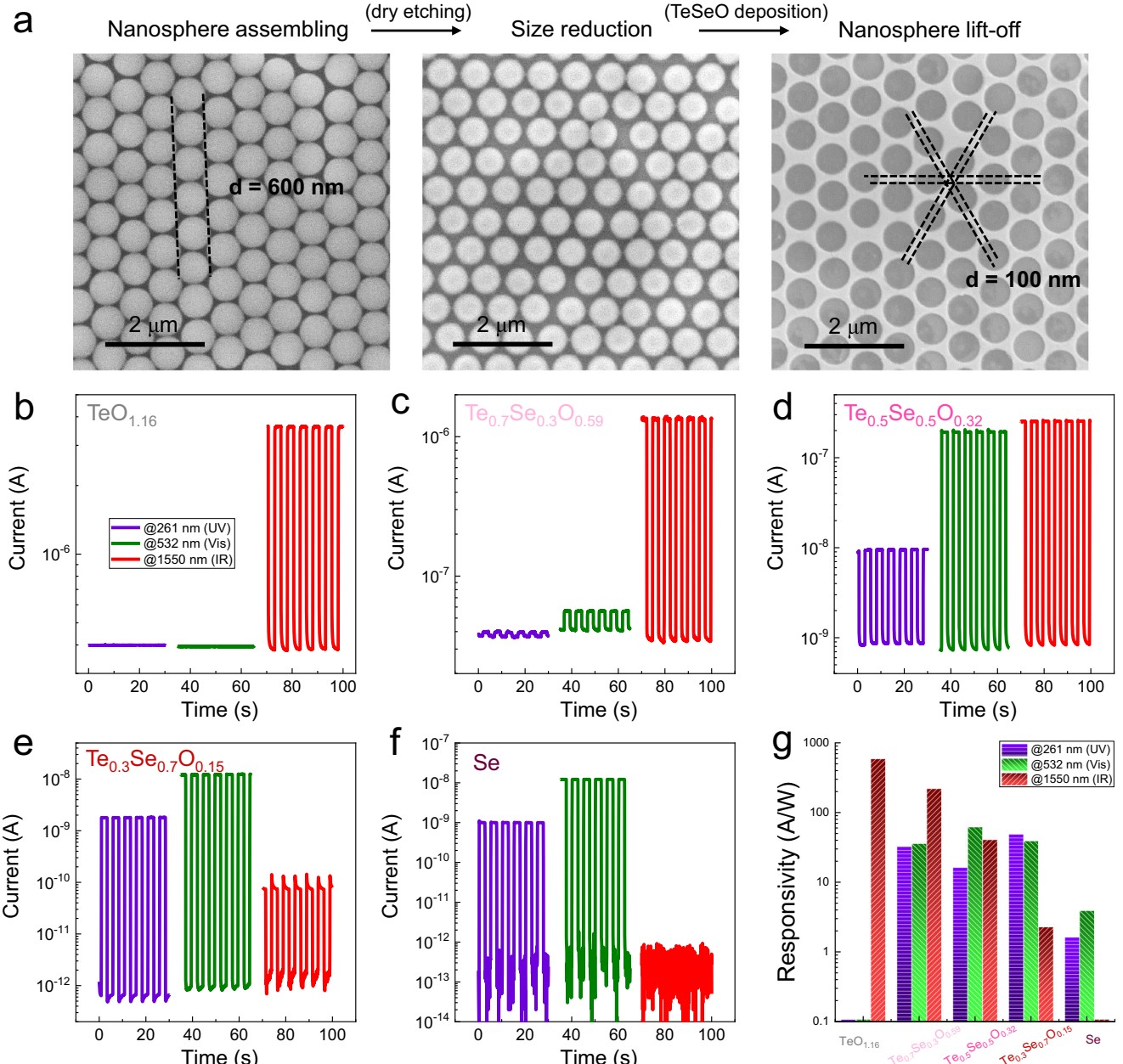

**Fig. 4 | Nanopatterned TeSeO and broadband photodetection. a** SEM images of the nanosphere lithography process consisting of nanosphere assembly, size reduction, TeSeO deposition, and nanosphere lift-off. **b**–**f** Broadband UV-vis-SWIR operation of flexible nanopatterned TeSeO photodetectors with a chopped frequency of 0.2 Hz. **g** Optical responsivity of TeSeO with different compositions under various incident light wavelengths.

stability under the long-term storage test (Supplementary Fig. 9). After 300 days of ambient storing, the TeSeO TFT performances show no discernible degradation in output current, $\mu_{FE}$, or hysteresis, even without device encapsulation. To the best of our knowledge, the superior operating/environmental durability of TeSeO is unachievable by other p-type thin-film counterparts[11].

## Broadband photodetection and mechanical robustness of nanopatterned TeSeO

Semiconducting nanostructures are promising for optoelectronics because of their high absorption coefficient and superior flexibility[10,27]. A maskless nanosphere lithography was employed as a low-cost sub-micron-scale structure fabrication method to produce honeycomb TeSeO nanostructures on flexible polyimide (PI) substrates (Fig. 4a and Supplementary Fig. 10, fabrication details shown in Method

section)[28–30]. To achieve a good trade-off between flexibility and conductivity, an inter-aperture wire width of ~100 nm was employed in both experimental study and theoretical modeling. After that, room-temperature photodetecting measurements were carried out using different UV (261 nm), visible (532 nm), and SWIR (1550 nm) light sources. The $Te_{0.7}Se_{0.3}O_{0.59}$, $Te_{0.5}Se_{0.5}O_{0.32}$, and $Te_{0.3}Se_{0.7}O_{0.15}$ samples show good photoresponse to these tested wavelengths and yield significant photocurrent under periodic illumination (Fig. 4c–e). Determined by their optical bandgap and absorption efficiency, weak SWIR response was observed at the Se-rich device (Fig. 4f), while the $TeO_{1.16}$ device is highly photosensitive to the SWIR (Fig. 4b). All the performance parameters of flexible TeSeO photodetectors (PDs) are calculated and summarized in Fig. 4g. Specifically, the responsivities of $TeO_{1.16}$ and $Te_{0.7}Se_{0.3}O_{0.59}$ under SWIR irradiation are 603 and 225 A/W, respectively, better than the reported intrinsic Te PDs and

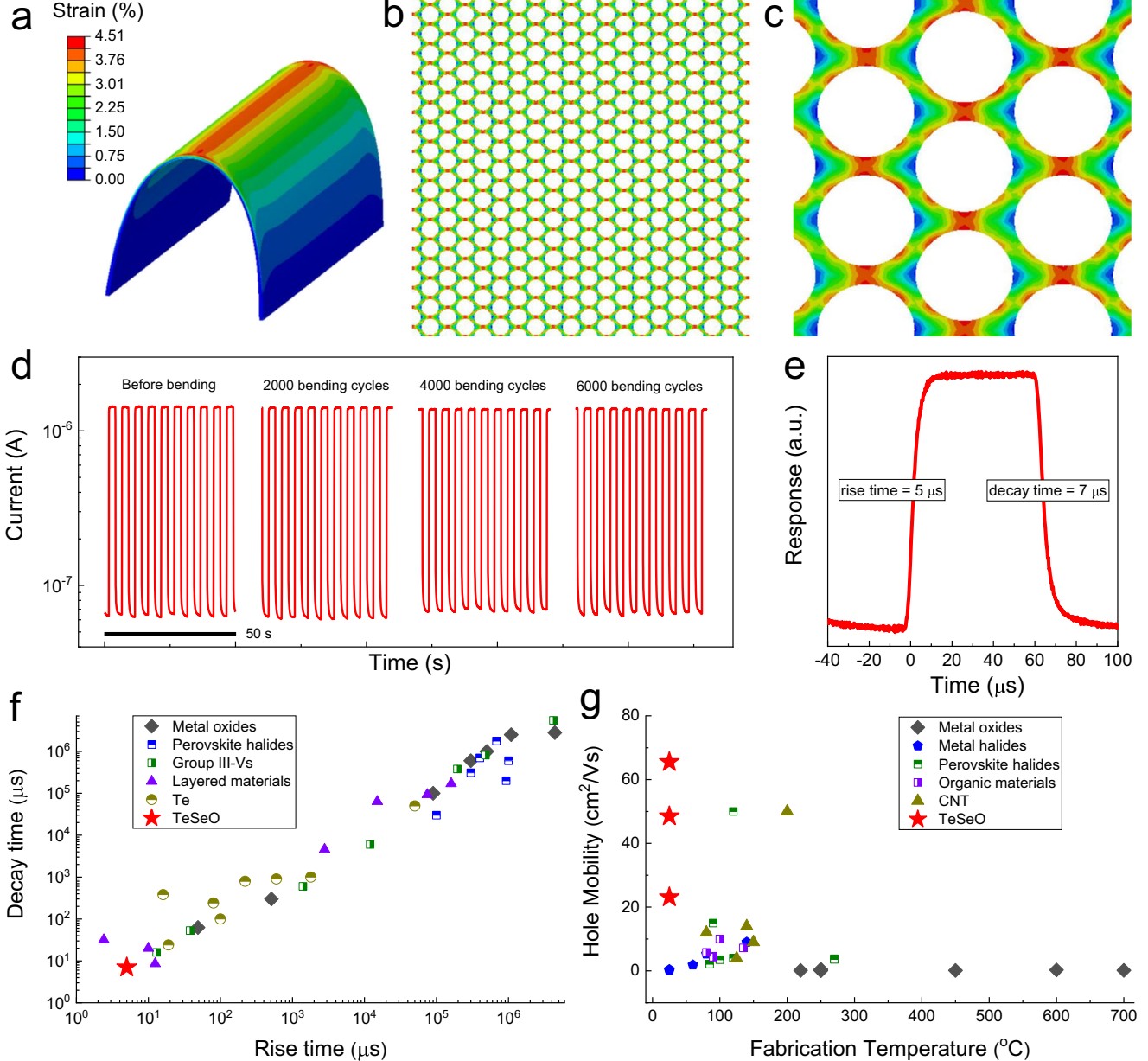

**Fig. 5 | Mechanical analysis of honeycomb TeSeO optoelectronics. a** FEA simulation of the TeSeO layer/PI substrate model at a bending radius of 1.5 mm. **b** Top views of strain distribution on the bent TeSeO honeycomb layer and **c** representative area. **d** The device photocurrent under on/off switching light illumination (0.2 Hz) as a function of bending cycles. **e** Time-resolved photoresponse of $Te_{0.7}Se_{0.3}O_{0.59}$ PDs measured at 1550 nm illumination with a chopped frequency of 10 kHz. Comparison of **f** photoresponse speed and **g** hole mobility of metal oxides, metal halides, perovskite halides, organic materials, TeSeO, etc. The corresponding references can be found in Supplementary Tables 3 and 4.

comparable to those state-of-the-art wideband PDs (Supplementary Table 4).

Honeycomb structures with nano/micro-scale geometric dimensions can accommodate mechanical deformations and thereby contribute to the superior flexibility of soft (opto-)electronics[31–33]. Here, the mechanical response of honeycomb TeSeO nanostructure subject to bending was evaluated by finite element analysis (FEA, see "Method" section). Impressively, benefiting from the porous structure, the strain on the TeSeO honeycomb channel located on the substrate center is efficiently dispersed with a bending radius of 1.5 mm (Fig. 5a–c and Supplementary Fig. 11)[34]. After that, the simulation conditions were fully reproduced in a real bending experiment to check the mechanical durability of nanopatterned TeSeO PDs directly. The whole bending test with bending times up to 6000 shows no detectable photocurrent

deterioration (Fig. 5d). In contrast, the TeSeO flat film without the nanopattern process displayed a significant resistance increase with the bending times (Supplementary Fig. 12), and eventually, the device broke down because of the appearance of micro-cracks after bending. Thus, it is indicated that the strain-induced structural damage (e.g., plastic deformation and crack initiation) and electrical deterioration in nanopatterned flexible TeSeO could be avoided effectively[35].

In addition to the high sensitivity and good mechanical flexibility demonstrated above, we benchmark our flexible PD performance with the transient response speed, which highly depends on the efficient collection/transport of photo-generated carriers[16,36,37]. The transient output signals of nanopatterned $Te_{0.7}Se_{0.3}O_{0.59}$ PDs were measured under modulated 1550 nm illumination. Even with a chopping frequency up to 10 kHz, the devices exhibit high response reliability

without signal distortion (Supplementary Fig. 13). More importantly, the $Te_{0.7}Se_{0.3}O_{0.59}$ devices exhibit ultra-fast optical response with the rise and decay times being 5 μs and 7 μs (Fig. 5e). The μs-level response time is better than most p-channel PDs reported in the literature (Fig. 5f), mainly due to the intrinsic high hole mobility of inorganic-blended TeSeO and the high surface-to-volume ratio of honeycomb structure (Fig. 5g). These observations promise future air-stable and high-speed optoelectronic applications based on p-type band-tunable semiconductors.

## Discussion

To summarize, TeSeO, a versatile p-type inorganic semiconductor system, was designed and deposited as thin films and honeycomb nanostructures at room temperature. By utilizing an inorganic blending strategy, the band structure of TeSeO was engineered to meet the specific technical requirements. For instance, by optimizing the TeSeO formulation, $Te_{0.8}Se_{0.2}O_{0.8}$ TFTs show high $\mu_{FE}$ of 48.5 cm²/(Vs) and $I_{on}/I_{off}$ of ~$10^5$, while the flexible honeycomb $Te_{0.7}Se_{0.3}O_{0.59}$ broadband PDs show fast optical response down to 5 μs. Importantly, benefiting from the partial oxidation in TeSeO, the devices exhibited good operational robustness under long-term storage and persistent bias. These performance parameters surpass those of conventional p-type thin films (e.g., metal oxides, metal halides, perovskite halides, and organic materials) and are on par with the state-of-art n-type scalable metal oxides (e.g., a-InGaZnO). In this regard, the inorganic-blended TeSeO system could be applied to diverse functional utilization beyond the immediate interests in (opto-)electronics.

## Methods

### Material synthesis

The whole fabrication process of TeSeO films was conducted at room temperature in a scalable manner. First, the $SiO_2$ and polyimide (PI) substrates used in this work were ultrasonically cleaned in acetone, ethanol, and deionized water and dried by nitrogen gas. Before using, Te (Sigma–Aldrich, 99.997%, powder) and Se (Sigma–Aldrich, 99.99%, powder) were mixed and then ground for 30 min. Due to the higher vapor pressure of Se than Te, less Se powder compared with the desired Te/Se ratio was added to the mixed source powder. For instance, to achieve the Te/Se ratios of 7/3, 5/5, and 3/7, the Se powder with percentages of 17%, 38%, and 55% was added to the mixed source, respectively. After that, a thermal evaporation process with a deposition rate of 2 Å/s was utilized to deposit TeSe thin films with a chamber pressure below $4 \times 10^{-6}$ Torr at room temperature. The film thickness is proportional to the deposition time, monitored by an INFICON SQC-310 deposition controller combined with a quartz crystal oscillator. To achieve oxygen implantation in TeSeO thin films, a standard oxygen implantation technique (PC-150, JunSun Tech Co., Ltd.) was employed with a plasma power of 30 W and an $O_2$ gas flow of 50 sccm. The chamber pressure was set to 0.26 Torr during the oxygen implantation process with a duration of 60 s.

### Material characterization

Surface morphologies of TeSeO films were examined with scanning electron microscopy (SEM, FEI Quanta 450 FEG SEM) and atomic force microscopy (AFM, Bruker Dimension Icon AFM). A Rigaku SmartLab X-ray Diffractometer (XRD) with Cu Kα radiation was used to evaluate the crystallinity and crystal structure of the TeSeO films. To get a stronger signal from the TeSeO film and avoid signal from the substrate, grazing-incidence XRD measurement was performed with a fixed grazing-incidence angle of 1°. Crystal structures were also determined by high-resolution transmission electron microscopy (HRTEM, JEOL 2100F). Elemental mappings were performed using an energy-dispersive X-ray spectroscopy (EDS) detector attached to a spherical-aberration-corrected scanning transmission electron microscopy (STEM, JEOL JEM-ARM300F2). To realize the elemental and chemical analysis of samples, a Thermo Scientific ESCALAB 250Xi system was employed to perform ultraviolet photoelectron spectroscopy (UPS) and X-ray photoelectron spectroscopy (XPS). Before UPS and XPS measurement, the samples were cleaned by $Ar^+$ ion etching to remove surface contamination. All the XPS peaks were calibrated by carbon (C 1$s$) peaked at 284.8 eV.

### Nanosphere lithography

The monodispersed suspension of polystyrene (PS) nanospheres (10 wt%, in water, diameter of 600 nm) was used in this work. First, PS nanospheres were self-assembled into close-packed hexagonal arrays at the water-air interface. Then, the close-packed monolayers of PS nanospheres were transferred onto a flexible PI substrate, serving as lithographic masks. After that, a time-dependent dry oxygen etching process was used to tailor the diameter of PS nanospheres to ~500 nm, which was performed under $O_2$ (50 sccm) at a pressure of 0.26 Torr and a radio frequency power of 30 W for 45 s. After TeSeO film deposition, the nanospheres were lifted off by ultrasonicating the samples in toluene for 60 s, and the TeSeO honeycomb layer with an inter-aperture wire width of ~100 nm was formed on the PI substrates.

### Finite element analysis (FEA)

The mechanical performance of honeycomb TeSeO nanostructures on flexible PI substrates under bending behavior was simulated and analyzed using FEA with the commercial software ABAQUS. The model used in this simulation was developed based on real bending experiments and incorporated the actual geometries and loading history. The PI substrate bent into a semi-circle with a radius of 1.5 mm, causing the attached honeycomb TeSeO nanostructures to deform accordingly. The slip between the substrate and the TeSeO film in the model is neglected. The linear elastic constitutive model was considered in the FEA simulation, where TeSeO has a Young's modulus of 31.1 GPa, and PI has a Young's modulus of 2.5 GPa, while their Poisson's ratios are 0.33 and 0.39, respectively. The maximum principle strain distribution of the TeSeO film was obtained in this model.

### Device fabrication and characterization

Bottom-gate top-contact thin-film transistors were constructed on $p^+$-Si/$SiO_2$ substrates, in which the thermally grown oxide thickness is 50 nm. Both channel layers and source/drain electrodes are patterned through shadow masking to guarantee low gate leakage currents and reliable parameter extraction. The TeSeO channel thickness is ~10 nm in this work. The 70-nm thick Ni source/drain electrodes were deposited by electron beam evaporation, with a channel width/length of 100 μm/40 μm. Ni electrodes have a high work function of ~5.1 eV, suitable for contact with p-type semiconductors. Agilent 4155 C semiconductor analyzer was employed to realize electrical characterizations with help from a standard electrical probe station. Field-effect mobility ($\mu_{FE}$) in the linear regime can be calculated using $\mu_{FE} = Lg_m/(WC_iV_{DS})$, where $L$, $W$, $C_i$, and $V_{DS}$ are the channel length, channel width, gate capacitance per unit area, drain-source voltage, respectively. The $g_m$ was transconductance defined as $\partial I_{DS}/\partial V_{GS}$, where $I_{DS}$ is drain-source current and $V_{GS}$ is gate-source voltage. The $C_i$ was calculated from the parallel plate capacitor model, using $C_i = (\varepsilon A)/d$, where the dielectric constant ($\varepsilon$) and film thickness ($d$) of the $SiO_2$ dielectric layer are 3.9 and 50 nm, respectively. The Ecopia HMS 5300 Hall effect measurement system, equipped with a 0.54 T permanent magnet, was employed to measure the carrier concentration and Hall mobility using the van der Pauw method. For the photodetector measurements, lasers with different wavelengths of ultraviolet (261 nm), visible (532 nm), and short-wave infrared (1550 nm) were used as the light sources, in which their incident light powers ($P$) were determined by a power meter (PM400, Thorlabs). To quantify the photodetecting performance, responsivity ($R$) was estimated using $R = I_p/(PA)$, where

$I_p$ is photocurrent (defined as light current minus dark current) and $A$ is the effective irradiated area. The rise and decay times of photodetectors are determined as the time to vary from 10% to 90% of the peak photocurrent and vice versa.

## Data availability

Relevant data supporting the key findings of this study are available within the article, the Supplementary Information file, and the Source Data file. All raw data generated during the current study are available from the corresponding authors. Source data are provided with this paper.

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

## Acknowledgements

This work is supported by a fellowship award from the Research Grants Council of the Hong Kong Special Administrative Region, China (CityURFS2021-1S04) and the Shenzhen Municipality Science and Technology Innovation Commission (grant no. SGDX2020110309300402; "Modulation and Detection of Terahertz Waves based on Semi-Metallic Two-Dimensional Materials," CityU). Y.L. and R.F. acknowledge the financial support from the Research Grants Council of Hong Kong SAR, China (Grant RFS2021-1S05), NSFC/RGC Joint Research Scheme (Grant N_HKU159/22) and the Science and Technology Department of Sichuan Province (Grant 2022YFSY0001).

## Author contributions

Y.M. and J.C.H. conceived and initiated the project. Y.M. prepared materials, fabricated devices, and performed analysis. R.F., X.C.L., and Y.L. performed the FEA simulations and microstructure analysis. W.J.W. and Z.X.L. performed the atomic force microscopy measurement and analysis. W.J.W. and W.W. contributed to the crystal structure study. Q.Q., W.J.W., P.S.X., D.C., Z.Y., and C.Y.W. performed part of analysis. D.J.L., H.S., B.W.L., Z.H.W., and S.P.Y. helped with electrical measurement and analysis. Y.M. and J.C.H. wrote the initial manuscript. All

authors contributed to the final manuscript and approved the submission.

## Competing interests

The authors declare no competing interests.
