## [Peer Review File · Nature Communications]

REVIEWER COMMENTS

Reviewer #3 (Remarks to the Author):

This work describes production and characterization of Te-Se-O thin films. Developing materials with continuously tunable electronic properties is useful, however I find there are important details and discussion missing from the paper which should be added before publication.

What is the film thickness, and how uniform are the films? Specifically, the expected oxygen content after implantation is not given, and it should be confirmed that the oxygen distribution is uniform. Based on the EDS in Figure S3, it appears that there may be less oxygen close to the SiO₂ substrate.

Are the films kept in an inert environment to protect from oxidation? Would a naturally forming oxide layer which may occur in higher Te concentrations influence results?

For the partially crystallized films, are the compositions of the crystallized vs amorphous regions similar?

Even for uniform composition, differences in microstructure e.g. grain sizes can influence transport properties. How can this effect be separated from the effects of the changing composition? For example, the data shown in Fig. 3(a-c) for Te_{0.7}Se_{0.3}O_x, Te_{0.8}Se_{0.2}O_x, and Te_{0.9}Se_{0.1}O_x were previously shown to have different amorphous-crystalline ratios, and likely very different grain sizes in the crystalline part. How do these changing microstructures influence the measurements, and what strategies could be used to control the microstructure?

Reviewer #4 (Remarks to the Author):

Meng et al has synthesized TeSeO_x films producing p-type oxide materials. The p-type oxide semiconductors are imperative in many applications in electronics and sensing and technologically very critical. I read the article with a great interest. The authors have produced high quality figures and well-written text as well. However, there are several concerns with the manuscript that authors need to address and at this point it is not suitable for nature communications journal. Concerns are as following:

1. There is a major Te (0) peak meaning that in all of the mix oxide the Te (0) is higher than Te (4+). The authors seem to be depositing substantial amount of Te (0).
2. XRD needs to also include TeO₂ reference powder as purchased matching literature to compare.
3. The band gap starts from less than 1 eV for Te oxide (mentioned TeO_x) and become wider by incorporating Se in Fig 2g however the table 1 earlier mentions TeO₂ with 3.7 eV and Te (0) with 0.31 eV again confirming authors are making Te (0) and not TeO_x. So this is a serious issue in the article.
4. authors mentioned O-Te-O has a higher binding (polarity) than Se-Te-O coordination and then the binding energy of Oxygen is increased when doping is increased meaning that the oxygen has a more localised valence electrons and therefore not supporting the fundamental discussion that the polarity is reduced. This is again another major concern.
5. Few of the SAED diffractions need to be indexes and amorphous rings need to be clearly labeled for identification and review.
6. thickness needs to be verified and measured
7. X in TeO_x and TeSeO_x need to be measured and shown.
8. The Sub-oxides of TeO_x location in XPS need to be clearly identified and are they expected to be different than Te (0)? It seems there is a mix usage of suboxide and Te(0).
9. The p-type mobility and carrier concentration need to be verified using hall mobility due to several inaccuracies found.
10. Fig S3 seem to have Si on Pt side is that due to etching process?

Reviewer #5 (Remarks to the Author):

This is an excellent paper introducing a successful design strategy for p-type Te-Se-O semiconductors fabricated at room temperature that are mechanically and chemically stable. The design basis is clearly explained, the structure of the films is fully examined, and the utility of the material in optoelectronic devices is demonstrated. In the end, the material displays high hole mobility and (for slightly different compositions) a rapid optical response time. I recommend publication in Nature Communications.

I do have a few very minor comments/corrections that the authors might address.

(1) On page 2, the phrase "...have been identified to process respectable..." should probably be "...have been identified to possess respectable..." or some such.

(2) When discussing the lattice parameters, the HRTEM data is quoted to only 2 digits of accuracy. Within this level of accuracy, the alloy lattice parameters are not slightly smaller than the Te quoted values, but rather are unchanged. (The comparison with the Se lattice parameters remains correct.)

(3) In the penultimate sentence of the paragraph beginning "In general,..." on page 4, the word "monotonously" should be replaced with "monotonically".

(4) In figure 5a (and in figure S11a) the key for the principal strain has the phrase (Avg 75%). What does this mean? Also, the peak strain in the model is nearly 4.5%. This is not an insignificant strain - for a typical material, one would see either plastic deformation or fracture before strains this large are observed. Nevertheless, the ligaments in the honeycomb structure appear to withstand these strains experimentally without much change in their optoelectronic response. Do the authors have an explanation for why their material can withstand such high stresses?

(5) I am a bit confused by the caption of Fig. S5. In particular, the final sentence discusses a parameter that "decreases" from 0.5 to 3.0. The parameter is Te/Te^{4+} . Should this be $\text{Te}^0/\text{Te}^{4+}$ and should the value be increasing?

Response to Reviewers' Comments

on Nature Communications Manuscript NCOMMS-23-35357-T

We appreciate the referees for their evaluation of our manuscript and for providing valuable feedback. Based on the reviewer's inputs, modifications highlighted in red have been made to the revised manuscript. A detailed account of these changes is presented in the point-by-point response provided below.

Reviewers' Comments:

Reviewer #3 (Remarks to the Author):

This work describes production and characterization of Te-Se-O thin films. Developing materials with continuously tunable electronic properties is useful, however I find there are important details and discussion missing from the paper which should be added before publication.

Reply:

We thank the referee for providing positive feedback on our manuscript. We recognize the significance of the comments on composition uniformity, surface oxidation, and microstructure control, and we are committed to enhancing the manuscript accordingly. Please refer to the point-by-point response below for a detailed explanation of the revisions made in response to the reviewer's comments.

What is the film thickness, and how uniform are the films? Specifically, the expected oxygen content after implantation is not given, and it should be confirmed that the oxygen distribution is uniform. Based on the EDS in Figure S3, it appears that there may be less oxygen close to the SiO₂ substrate.

Reply:

We extend our gratitude to the referee for your valuable comments. Unless explicitly specified, it is imperative to note that most characterizations and device constructions were executed on TeSeO films with a thickness of approximately 10 nm. The TeSeO thickness of ~10 nm was carefully selected for its ability to fulfill the requirements of both current driving and gating regulation. The original Figure S3 showcased a cross-sectional TeSeO sample with a considerably larger film thickness exceeding 100 nm. This thickness may result in insufficient oxygen implantation at the bottom of the sample, as evidenced by the diminished oxygen distribution in proximity to the SiO₂ substrate. Conducting EDS mapping on such a thick TeSeO film is a careless mistake, which cannot accurately reflect the elemental distributions in a useful depth. In the revised Supplementary Information, the EDS mapping of the 10-nm-thick TeSeO film was updated to Figure S3 to show the elemental distributions better.

Accordingly, the sentence "The channel thickness of ~10 nm, which is verified by cross-sectional TEM, is used to balance the conductivity and the on/off current ratio." was modified on page 5 of the revised manuscript.

In response to the comment, we have provided a comprehensive analysis in the revised manuscript focusing on TeSeO films with a standard thickness of 10 nm. Multiple assessments, encompassing composition, morphology, and device performance, have been included to underscore the uniformity and smoothness of these semiconducting thin films. These analyses collectively reinforce the practical viability of the presented device utilizations.

1) Composition uniformity. To assess the elemental homogeneity of the film, we employed X-ray photoelectron spectroscopy (XPS) characterization on ten distinct positions of TeSeO thin films deposited on a 4-inch SiO₂/Si substrate. The wafer-scale composition uniformity is substantiated by the XPS analysis of Te 3*d*, O 1*s*, and Se 3*d*, revealing consistent shapes, areas, and peak positions in the spectra obtained from different film positions. The corresponding XPS results were added to Figure S4 on page 5 of the revised Supplementary Information (also shown below).

Figure S4. XPS (a) Te 3*d*, (b) O 1*s*, and (c) Se 3*d* analysis of ten different positions of TeSeO thin films grown on a 4-inch SiO₂/Si wafer.

2) Morphological uniformity. Surface morphologies of TeSeO thin films with varying composition ratios were scrutinized using atomic force microscopy (AFM, Bruker Dimension Icon AFM), covering a scanning area of 10×10 μm (Figure 1c). All TeSeO thin films exhibit a smooth, uniform, and crack-free surface. The extracted arithmetic mean deviation of roughness

shows a consistent decrease from 1.8 nm to 0.3 nm with an increase in Se content, as illustrated in the inset of Figure 1c.

3) Device performance uniformity. To assess the scalability and uniformity of TeSeO thin-film transistors (TFTs), we fabricated wafer-scale TFT arrays (10×10 array), and the statistical distribution of device performance is illustrated in Figure S7. The TFT array shows 100% device yield with a hole mobility of $48.2 \pm 8.4 \text{ cm}^2/(\text{Vs})$, $I_{\text{on}}/I_{\text{off}}$ of $10^4 \sim 10^5$, and V_{TH} of $-4.2 \pm 1.3 \text{ V}$.

Besides, following the reviewer's suggestion, the oxygen content after implantation was given in the revised manuscript. To ascertain the composition ratios of TeSeO thin films, we conducted a comprehensive analysis using a combination of energy-dispersive X-ray spectroscopy (EDS) mapping and XPS, as depicted in Figures S3 to S5 of the revised Supplementary Information. The accurate composition ratios of TeSeO compounds were determined (e.g., $\text{TeO}_{1.16}$, $\text{Te}_{0.7}\text{Se}_{0.3}\text{O}_{0.59}$, and $\text{Te}_{0.5}\text{Se}_{0.5}\text{O}_{0.32}$), and the corresponding mole ratios of Te^0 and oxidized Te^{4+} are presented in Table S1 on page 7 of the revised Supplementary Information (also shown below).

Table S1. Summary of composition ratios of TeSeO films and mole ratios between Te^0 and Te^{4+} .

Sample compositions	Mole ratios (mol%)	
	Te^0	Te^{4+}
$\text{TeO}_{1.16}$	42	58
$\text{Te}_{0.9}\text{Se}_{0.1}\text{O}_{0.98}$	45	55
$\text{Te}_{0.8}\text{Se}_{0.2}\text{O}_{0.80}$	50	50
$\text{Te}_{0.7}\text{Se}_{0.3}\text{O}_{0.59}$	58	42
$\text{Te}_{0.6}\text{Se}_{0.3}\text{O}_{0.44}$	63	37
$\text{Te}_{0.5}\text{Se}_{0.5}\text{O}_{0.32}$	68	32
$\text{Te}_{0.4}\text{Se}_{0.6}\text{O}_{0.23}$	71	29
$\text{Te}_{0.3}\text{Se}_{0.7}\text{O}_{0.15}$	75	25
$\text{Te}_{0.2}\text{Se}_{0.8}\text{O}_{0.09}$	77	23
$\text{Te}_{0.1}\text{Se}_{0.9}\text{O}_{0.04}$	80	20
Se	0	0

Are the films kept in an inert environment to protect from oxidation? Would a naturally forming oxide layer which may occur in higher Te concentrations influence results?

Reply:

We appreciate the insightful comment and agree with the possibility of a naturally forming partly oxide layer, especially in samples with higher Te concentrations. While ultrathin Te-based films are more air-stable than certain elemental materials such as black phosphorus, they are susceptible to oxidation when stored in the ambient. Indeed, our XPS results reveal that approximately 10% of neutral Te undergoes oxidation to the Te^{4+} state when $\text{Te}_{0.7}\text{Se}_{0.3}$ films are

exposed to the ambient environment, as shown in Figure R1 below. This oxidation of neutral Te to Te^{4+} occurs during deposition and/or exposure to the ambient atmosphere, a phenomenon also reported in other studies. For instance, Smyth et al. reported a similar uncontrolled surface oxidation process in Te films within just 10 minutes of exposure to the ambient environment.¹

In our work, we employed an oxygen implantation process to enhance the oxidized Te content, defined as $\text{Te}^{4+}/(\text{Te}^0 + \text{Te}^{4+})$, reaching levels such as 58% for $\text{TeO}_{1.16}$ and 42% for $\text{Te}_{0.7}\text{Se}_{0.3}\text{O}_{0.59}$. After applying oxygen implantation to TeSeO thin films, the intentionally introduced oxygen serves to impede the uncontrolled formation of a surface oxide layer. This oxidation resistance is reflected in the stable operational stability observed under ambient storage conditions. Thanks to the partial oxidation in TeSeO thin films, which acts as a barrier against environmental influences, the devices demonstrated consistent operational stability during a long-term storage test (see Figure S9). Notably, after 300 days of ambient storage, the TeSeO TFT performances exhibited no discernible degradation in the output current, μFE , or hysteresis, even without device encapsulation.

Figure R1. XPS Te 3d analysis of (a) $\text{Te}_{0.7}\text{Se}_{0.3}$ film and (b) $\text{Te}_{0.7}\text{Se}_{0.3}\text{O}_{0.59}$ film.

For the partially crystallized films, are the compositions of the crystallized vs amorphous regions similar?

Reply:

We thank the referee for raising this question. As illustrated in Figure 1e, the selected area electron diffraction (SAED) pattern of $\text{Te}_{0.7}\text{Se}_{0.3}\text{O}_{0.59}$ exhibits diffraction spots corresponding to the hexagonal TeSe phase, while the characteristic diffuse halo is associated with amorphous regions. Additionally, XPS spectra indicate the presence of high-content Te^{4+} bonding and O-Te-O bonding, whereas Raman and XRD patterns reveal no O-related diffraction peaks. Collectively, these characterization results suggest the existence of an amorphous state of oxides in TeSeO. Similarly, Kim et al. also observed the local amorphous oxidized phase between hexagonal Te crystal domains.²

As mentioned before, the wafer-scale multiple-position XPS analysis confirms the element homogeneity at a macroscopic level. To delve into the local composition differences between crystallized and amorphous regions in the TeSeO film, we conducted EDS mapping using a spherical-aberration-corrected scanning transmission electron microscopy (STEM, JEOL JEM-

ARM300F2) from a cross-sectional view. The STEM high-angle annular dark field (HAADF) image shows good quality TeSeO film with a thickness of 10 nm, consisting of crystallized and amorphous regions. From a microscale, the distributions of Te, Se, and O signals are relatively uniform without noticeable element segregation. Considering that EDS mapping is a 2D mapping tool, the possible differences in compositions located at crystallized and amorphous regions may be overshadowed. In this regard, 3D quantitative chemical analysis with near-atomic resolution, e.g., 3D atom probe tomography (3D-APT), would be a useful tool to access reliable information on local compositions in the future.

Figure S3. (a) Cross-sectional STEM-HAADF image and (b) EDS mapping of TeSeO film.

Accordingly, the cross-sectional STEM-HAADF image and EDS mappings of 10-nm-thick TeSeO film were shown in Figure S3 on page 4 of the revised Supplementary Information.

Even for uniform composition, differences in microstructure e.g., grain sizes can influence transport properties. How can this effect be separated from the effects of the changing composition? For example, the data shown in Fig. 3(a-c) for $Te_{0.7}Se_{0.3}O_x$, $Te_{0.8}Se_{0.2}O_x$, and $Te_{0.9}Se_{0.1}O_x$ were previously shown to have different amorphous-crystalline ratios, and likely very different grain sizes in the crystalline part. How do these changing microstructures influence the measurements, and what strategies could be used to control the microstructure?

Reply:

We appreciate the valuable comments from the reviewer, acknowledging the influence of both composition and microstructure on the transport properties of TeSeO films. Over decades, intensive research has established that many factors, such as substrate temperature, deposition rate, and the introduction of nucleation sites, can effectively alter grain sizes of T-based materials, thereby influencing transport properties.^{3,4} In our study, we conducted a control experiment with different deposition rates in the physical vapor deposition (PVD) process, using rates of 2 Å/second, 1 Å/second, and 0.5 Å/second, while keeping other parameters

constant, including chamber pressure ($\sim 4 \times 10^{-6}$ Torr), room-temperature deposition, and SiO₂/Si substrates.

To assess the intrinsic electrical properties of different TeSeO samples, we performed Hall measurements using the van der Pauw method on Te_{0.8}Se_{0.2}O_{0.8} samples at room temperature. Interestingly, the Te_{0.8}Se_{0.2}O_{0.8} films deposited at different deposition rates (including 2 Å/second, 1 Å/second, and 0.5 Å/second) exhibited a similar hole density of around 8×10^{17} cm⁻³, as presented in Table R1 (also shown below). This result may relate to the compensating effect of structural defects induced by higher deposition rates.⁵ With different TeSeO compositions deposited at the same deposition rates of 2 Å/second, the hole carrier concentrations are varied, showing 4.0×10^{17} cm⁻³, 8.2×10^{17} cm⁻³, and 1.1×10^{18} cm⁻³, for Te_{0.7}Se_{0.3}O_{0.59}, Te_{0.8}Se_{0.2}O_{0.8}, and Te_{0.9}Se_{0.1}O_{0.98}, respectively, as presented in Table S2 on page 9 of the revised Supporting Information (also shown below).

Notably, higher deposition rates correlated with higher average Hall mobility values, such as 30.8 cm²/(Vs) for samples deposited at 0.5 Å/second and 45.2 cm²/(Vs) for samples deposited at 2 Å/second, as presented in Table R1. Hall mobility, primarily determined by the scattering time interval during charge-carrier transport, is influenced by factors like crystal structure disorders and the density of grain boundaries.⁶ Hence, the higher deposition rates effectively reduce grain boundary density and suppress charge-carrier scattering, providing a rationale for the enhanced Hall mobility values.

Table R1. Summary of the Hall mobility and hole concentration of Te_{0.8}Se_{0.2}O_{0.8} thin films with different deposition rates.

Deposition rates	Hall mobility (cm ² /Vs)	hole concentration (cm ⁻³)
2 Å/second	45.2	8.2×10^{17}
1 Å/second	35.9	7.9×10^{17}
0.5 Å/second	30.8	8.1×10^{17}

Table S2. Summary of the Hall mobilities and hole concentrations of TeSeO thin films.

Material	Hall mobility (cm ² /Vs)	hole concentration (cm ⁻³)
Te_{0.9}Se_{0.1}O_{0.98}	60.8	1.1×10^{18}
Te_{0.8}Se_{0.2}O_{0.8}	45.2	8.2×10^{17}
Te_{0.7}Se_{0.3}O_{0.59}	20.5	4.0×10^{17}

These results give a preliminary understanding of TeSeO electrical properties: carrier concentrations are influenced by composition, while mobility values are mainly regulated by microstructure, particularly grain sizes. This result is a preliminary finding, and more in-depth investigations on TeSeO with advanced tools are warranted.

Reviewer #4 (Remarks to the Author):

Meng et al has synthesized TeSeO_x films producing p-type oxide materials. The p-type oxide semiconductors are imperative in many applications in electronics and sensing and technologically very critical. I read the article with a great interest. The authors have produced high quality figures and well-written text as well. However, there are several concerns with the manuscript that authors need to address and at this point it is not suitable for nature communications journal. Concerns are as following:

Reply:

We extend our appreciation to the referee for the valuable comments provided. The feedback concerning atomic coordination, composition, and electrical properties is constructive and instrumental in refining the manuscript. Please refer to the point-by-point response below for a detailed overview of the revisions made in response to the comments.

1. There is a major Te (0) peak meaning that in all of the mix oxide the Te (0) is higher than Te (4+). The authors seem to be depositing substantial amount of Te (0).

Reply:

We appreciate the valuable comments from the reviewer. The coexistence of Te⁴⁺ and Te⁰ in the TeSeO thin films plays a key role in optimizing their material properties. We are sorry for not providing a detailed description of the partial oxidation of Te in TeSeO materials in our original manuscript, which led to a misunderstanding. From the XPS Te 3d spectra of TeSeO (see Figure 2a), the Te⁴⁺/(Te⁰+Te⁴⁺) ratios were determined to be 58%, 42%, 32%, and 25% for TeO_{1.16}, Te_{0.7}Se_{0.3}O_{0.59}, Te_{0.5}Se_{0.5}O_{0.32}, and Te_{0.3}Se_{0.7}O_{0.15}, respectively. The reduced Te⁴⁺/(Te⁰+Te⁴⁺) ratios with increasing Se content (Figure S5) indicate that the addition of Se suppresses the binding process between Te and O. This Se-regulated Te partial oxidation offers a means to shift the material compositions from Te⁴⁺-rich to Te⁰-rich. Given that TeO₂ is a p-type wide-bandgap semiconductor and Te is a p-type semimetal, altering the composition of TeSeO can significantly modify their corresponding physical/chemical properties across a wide range. Particularly, semimetal Te, with its high intrinsic hole mobility (~10⁵ cm²/Vs) and small effective hole mass (~0.3 m₀), predominantly contributes to the excellent hole-transport property of TeSeO.

It is crucial to note that while TeO₂ is considered a promising p-type wide-bandgap semiconductor, our previous experiments with tellurium dioxides (including pure TeO₂ and O-deficient TeO_{2-x} samples) revealed high resistance and no transistor behavior. Kim et al. also noted a similar finding: when tellurium oxides were added to the Te film by oxygen reactive sputtering, the resulting TFTs would degrade its on-current and lose their p-type transistor behaviors.⁶ From our findings and reported works, modifications to TeO₂-based semiconducting materials are essential to meet the requirements of device construction. Consequently, we designed the inorganic blending TeSeO system, strategically combining intrinsic p-type semimetal, semiconductor, and wide-bandgap semiconductor into a single compound. This inorganic blending strategy imparts advantages to p-type inorganic semiconductors, facilitating the construction of high-performance and cost-effective p-channel devices:

1) The TeSeO system achieves wide bandgap tunability from 0.7 eV to 2.2 eV, covering the ultraviolet, visible, and short-wave infrared wavelength regions. This characteristic endows TeSeO with high mobility, broadband photodetection ability, and a fast μs -level photoresponse.

2) Wafer-scale ultrathin TeSeO films exhibit a high hole field-effect mobility of $48.5 \text{ cm}^2/(\text{Vs})$ and robust hole transport properties. This performance is attributed to the coexistence of Te-Te (Se) and O-Te-O portions within the material.

3) Incorporating oxygen implantation results in the formation of O-Te-O (polar covalent bonds) with a wider bandgap. This feature imparts superior operating durability to TeSeO, a quality not achievable by other p-type thin-film semiconductors, including metal oxides, metal halides, and organic materials.

Accordingly, in response to the referee's comments, the quantitative chemical analysis, including composition ratios of TeSeO films and mole ratios between Te^0 and Te^{4+} , was added to Figures S3 to S5 of the revised Supporting Information. Also, the TeSeO compositions with accurate stoichiometric ratios were fully revised in the revised manuscript and Supplementary Information, and the sentence "**The corresponding $\text{Te}^{4+}/(\text{Te}^{0+}+\text{Te}^{4+})$ ratios decrease from $\sim 58\%$ ($\text{TeO}_{1.16}$) to $\sim 25\%$ ($\text{Te}_{0.3}\text{Se}_{0.7}\text{O}_{0.15}$) with increasing Se content (Figure S5)**" was modified on page 4 of the revised manuscript.

2. XRD needs to also include TeO_2 reference powder as purchased matching literature to compare.

Reply:

We appreciate your valuable input. In response to the referee's suggestion, TeO_2 reference peaks were incorporated into the XRD analysis (also shown below). The TeO_2 reference peaks were measured using purchased TeO_2 powder (Tellurium dioxide, 99.995%, Sigma Aldrich), which is known to be thermodynamically stable and readily available. All the powder XRD peaks are found to align well with the tetragonal TeO_2 with space group P41212 (92), which exhibits a distorted rutile structure with asymmetric covalent O-Te-O bonds.

For our TeSeO samples, no O-related diffraction peaks were detected in the XRD study. At the same time, high-content Te^{4+} bonding and O-Te-O bonding were observed in the XPS spectra. These results collectively suggest an amorphous state of oxides in TeSeO. Similarly, several research groups also reported the observation of amorphous oxide phases in Te-based thin films, in which ambient thermal annealing or oxygen-reactive sputtering was employed to introduce oxygen to the Te-based films.^{2,6,7}

Figure S1. GIXRD patterns of TeSeO thin films with different compositions and XRD pattern of the α -TeO₂ powder.

Accordingly, XRD pattern of α -TeO₂ powder was added to Figure S1 on page 2 of the revised Supplementary Information.

3. The band gap starts from less than 1 eV for Te oxide (mentioned TeO_x) and become wider by incorporating Se in Fig 2g however the table 1 earlier mentions TeO₂ with 3.7 eV and Te (0) with 0.31 eV again confirming authors are making Te (0) and not TeO_x. So this is a serious issue in the article.

Reply:

We appreciate the referee's insightful comment. It seems there might be a misunderstanding due to unclear expression in our manuscript. In this work, the tunable bandgaps of the TeSeO system are attributed to the synergistic effect of Se substitution and partial oxidation of Te. Both of these effects play crucial roles in our TeSeO inorganic blended system. As a strong glass former, the addition of Se contents would disturb the crystallization process of TeSeO, leading to the transition of the polycrystalline-to-amorphous phase. The Se-regulated Te oxidation allows for a variation in material compositions from Te⁴⁺-rich to Te⁰-rich, significantly modifying their corresponding physical/chemical properties across a wide range. The introduction of oxygen through implantation forms O-Te-O with a wider bandgap, providing superior operating durability to TeSeO that cannot be achieved by other p-type thin-film semiconductors.

The Te 3d spectra, as depicted in Figure 2a, clearly show the coexistence of Te⁴⁺ and Te⁰ peaks in the TeSeO thin films. This direct observation indicates the partial oxidation of Te in the TeSeO films. The partial oxidation of Te, as well as the Se addition, leads to a bandgap value larger than that of pure Te (0.31 eV) and smaller than that of TeO₂ (3.7 eV), showing wide bandgap tunability of TeSeO from 0.7 eV to 2.2 eV. This result agrees with the previously reported tellurium oxides with a bandgap of 0.9 eV (mentioned as TeO_x in that paper with a

Te⁴⁺/(Te⁰+Te⁴⁺) ratio of 31%),⁷ and the partially oxidized Te with bandgaps ranging from 0.7 to 2.8 eV depending on the oxygen flow rates during reactive magnetron sputtering.⁶ To avoid confusion, the accurate composition ratios of TeSeO were determined by using a combination of XPS and EDS, and the corresponding mole ratios of Te⁰ and oxidized Te⁴⁺ are presented in Table S1 on page 7 of the revised Supplementary Information (also shown below).

Table S1. Summary of composition ratios of TeSeO films and mole ratios between Te⁰ and Te⁴⁺.

Sample compositions	Mole ratios (mol%)	
	Te ⁰	Te ⁴⁺
TeO _{1.16}	42	58
Te _{0.9} Se _{0.1} O _{0.98}	45	55
Te _{0.8} Se _{0.2} O _{0.80}	50	50
Te _{0.7} Se _{0.3} O _{0.59}	58	42
Te _{0.6} Se _{0.3} O _{0.44}	63	37
Te _{0.5} Se _{0.5} O _{0.32}	68	32
Te _{0.4} Se _{0.6} O _{0.23}	71	29
Te _{0.3} Se _{0.7} O _{0.15}	75	25
Te _{0.2} Se _{0.8} O _{0.09}	77	23
Te _{0.1} Se _{0.9} O _{0.04}	80	20
Se	0	0

4. Authors mentioned O-Te-O has a higher binding (polarity) than Se-Te-O coordination and then the binding energy of Oxygen is increased when doping is increased meaning that the oxygen has a more localised valance electrons and therefore not supporting the fundamental discussion that the polarity is reduced. This is again another major concern.

Reply:

We appreciate the referee's insightful comment. We agree with the referee that, in general terms, as the binding polarity is enhanced, the valence band tends to become more localized. Also, it is important to note that this relationship is not always straightforward, and other factors, such as crystal structure and electron-electron interactions, can also play a role.

1) Crystal structure influence. The arrangement of atoms in the crystal lattice can significantly impact the binding strength and energy band structure.⁸ In our work, the TeSeO experienced significant crystal structure evolution with changing compositions. As shown in Figure 1b, the TeSeO samples with relatively high Te content (Te:Se \geq 7:3) show a polycrystalline nature, while the Se-rich samples (Te:Se \leq 6:4) are found to be amorphous. All the diffraction peak positions slightly shift to higher angles with increasing Se content (Figure S1), indicating the decrease of the lattice constant. At the same time, the increased full width at half-maximum of the diffraction peaks also reveals the suppressed material crystallinity. All these significant differences in crystal structures might lead to varying degrees of localization of the valence band.⁹

2) Electron-electron interactions. Interactions between electrons can influence the electronic structure of solid-state semiconductors. Strong electron-electron correlations can lead to deviations from the expected behavior based solely on the type of bonding. In general, the injection/extraction of electrons into/from a semiconductor will cause a downshift/upshift of the energy bands, corresponding to an increase/decrease of binding energy.^{10,11} As shown in Figures 2a-c, all the XPS core energy levels of TeSeO exhibit redshifts of up to 800 meV with increasing Se content, resulting from the electron injection process.¹⁰ This trend aligns well with the energy structure analyses and is further supported by Hall effect measurements, where the average hole carrier concentrations decrease from $1.1 \times 10^{18} \text{ cm}^{-3}$ ($\text{Te}_{0.9}\text{Se}_{0.1}\text{O}_{0.98}$) to $4.0 \times 10^{17} \text{ cm}^{-3}$ ($\text{Te}_{0.7}\text{Se}_{0.3}\text{O}_{0.59}$).

Moreover, TeSeO materials exhibit mixed bonding characteristics, in which the O-Te-O bond represents a polar covalent bond with a wider bandgap. In contrast, the Se-Te (or Te-Te) bond is a covalent bond featured with a semiconducting (or semimetal) nature. All these material-specific characteristics will influence the relationship between binding polarity and valence electron localization in our inorganic blended TeSeO compound system.

Accordingly, the sentence "All three XPS core energy levels of TeSeO exhibit redshifts of up to 800 meV with increasing Se content, resulting from the electron injection process." was modified on page 3 of the revised manuscript.

5. Few of the SAED diffractions need to be indexes and amorphous rings need to be clearly labeled for identification and review.

Reply:

We appreciate the critical comment. Following the referee's suggestion, SAED diffractions were indexed, and amorphous rings were labeled in Figure 1e of the revised manuscript (also shown below). The SAED patterns of $\text{TeO}_{1.16}$ and $\text{Te}_{0.7}\text{Se}_{0.3}\text{O}_{0.59}$ display a few diffraction spots, further indicating their polycrystalline structure. The observed diffuse halo, particularly in $\text{Te}_{0.5}\text{Se}_{0.5}\text{O}_{0.32}$, indicates that the addition of Se induces a polycrystalline-to-amorphous phase transition in TeSeO.

Figure 1 (e) SAED patterns of $\text{TeO}_{1.16}$, $\text{Te}_{0.7}\text{Se}_{0.3}\text{O}_{0.59}$, and $\text{Te}_{0.5}\text{Se}_{0.5}\text{O}_{0.32}$ thin films.

Accordingly, the sentence "The observed characteristic diffuse halo, particularly evident in $\text{Te}_{0.5}\text{Se}_{0.5}\text{O}_{0.32}$, indicates that the addition of Se induces a polycrystalline-to-amorphous phase transition in TeSeO." was modified on page 3 of the revised manuscript.

6. Thickness needs to be verified and measured

Reply:

We thank the referee for the valuable input. Film thickness calibration is crucial for Te-based functional films, given their well-known thickness-dependent electrical characteristics and the tradeoff between current driving and gating regulation abilities.¹² In our physical vapor deposition (PVD) process, the film thickness is monitored using an INFICON SQC-310 deposition controller equipped with a quartz crystal oscillator. Additionally, we performed calibration using atomic force microscopy (AFM) to ensure accuracy.

Here, responding to the reviewer's suggestion, we further validated the film thickness of TeSeO through cross-sectional scanning transmission electron microscopy (STEM) from a cross-sectional view (also shown below). The STEM high-angle annular dark field (HAADF) image confirms a channel thickness of approximately 10 nm, aligning well with the measurement obtained from the deposition controller/quartz crystal oscillator as well as AFM. This consistency underscores the accuracy and reliability of our film thickness monitor and calibration methods.

Figure S3. (a) Cross-sectional STEM-HAADF image and (b) EDS mapping of TeSeO film.

Accordingly, the sentence "The channel thickness of ~10 nm, which is verified by cross-sectional STEM, is used to balance conductivity and the on/off current ratio." was modified on page 5 of the revised manuscript, while the cross-sectional STEM-HAADF image of 10-nm-thick TeSeO film was shown in Figure S3 on page 4 of the revised Supplementary Information.

7. X in TeO_x and $TeSeO_x$ need to be measured and shown.

Reply:

We appreciate the referee for the valuable comment. In response to the suggestion, the composition ratios of TeSeO thin films were thoroughly examined using a combination of EDS and XPS. The accurate composition ratios of TeSeO were determined (e.g., $TeO_{1.16}$, $Te_{0.7}Se_{0.3}O_{0.59}$, and $Te_{0.5}Se_{0.5}O_{0.32}$), and the corresponding mole ratios of Te^0 and oxidized Te^{4+} were presented in Table S1 on page 7 of the revised Supporting Information (more detailed information could be found in the response to question 3).

8. The Sub-oxides of TeO_x location in XPS need to be clearly identified and are they expected to be different than $Te(0)$? It seems there is a mix usage of suboxide and $Te(0)$.

Reply:

We thank the important comment. We are sorry for this unclear way of expression about Te 3d peaks in the original manuscript. To avoid the misunderstanding, the revised manuscript includes a more explicit presentation of the XPS spectra of Te 3d fitting, where Te^0 and Te^{4+} peaks are clearly distinguished at ~ 572.8 eV and ~ 576.0 eV, respectively (also shown below). It is emphasized that the low-valence Te, such as Te^{2+} and Te^{z+} ($0 < z < 2$), and oxygen vacancies (V_O) are nearly undetectable in the TeSeO of this study.^{7,13} In our work, we employed an oxygen implantation process to enhance the oxidized Te content, defined as $Te^{4+}/(Te^{0+}Te^{4+})$, reaching levels such as 58% for $TeO_{1.16}$ and 42% for $Te_{0.7}Se_{0.3}O_{0.59}$. The efficient oxygen implantation in TeSeO of this work is considered a key factor in avoiding the generation of such metastable chemical states.

Figure S5. XPS Te 3d analysis of (a) $TeO_{1.16}$, (b) $Te_{0.7}Se_{0.3}O_{0.59}$, (c) $Te_{0.5}Se_{0.5}O_{0.32}$, and (d) $Te_{0.3}Se_{0.7}O_{0.15}$ thin films.

Accordingly, the XPS Te 3d analysis was added to Figure S5 on page 6 of the revised Supporting Information. The sentences "The coexisted Te⁴⁺ and Te⁰ peaks are clearly distinguished at ~572.8 eV and ~576.0 eV, respectively, which means the partial oxidation of Te. With increasing Se content, the corresponding Te⁴⁺/(Te⁰+Te⁴⁺) ratios decrease from ~58% (TeO_{1.16}) to ~25% (Te_{0.3}Se_{0.7}O_{0.59})" were added to the caption of Figure S5 on page 6 of the revised Supporting Information.

9. The p-type mobility and carrier concentration need to be verified using hall mobility due to several inaccuracies found.

Reply:

We appreciate the referee for the valuable comment. In response to the suggestion, we performed Hall effect measurements to verify the Hall mobility and carrier concentration of TeSeO films. The Hall measurement samples were prepared for Te_{0.7}Se_{0.3}O_{0.59}, Te_{0.8}Se_{0.2}O_{0.8}, and Te_{0.9}Se_{0.1}O_{0.98} films fabricated in a square configuration with an edge length of 10 mm. The Ecopia HMS 5300 Hall effect measurement system, equipped with a 0.54 T permanent magnet, was employed to measure the carrier concentration and Hall mobility using the van der Pauw method. To ensure the accuracy, ten measurements were conducted for each TeSeO sample, and the averages were calculated.

As a result, the Te_{0.7}Se_{0.3}O_{0.59} film shows an average hole carrier concentration of 4.0×10^{17} cm⁻³, which increases to 8.2×10^{17} cm⁻³ for Te_{0.8}Se_{0.2}O_{0.8} and 1.1×10^{18} cm⁻³ for Te_{0.9}Se_{0.1}O_{0.98} films (also shown below). This trend corresponds to the increasing density of hole generators, aligning well with the energy structure analyses. Correspondingly, the average Hall mobility values of the TeSeO films increase from 20.5 cm²/(Vs) for Te_{0.7}Se_{0.3}O_{0.59} to 45.2 cm²/(Vs) for Te_{0.8}Se_{0.2}O_{0.8} and further to 60.8 cm²/(Vs) for Te_{0.9}Se_{0.1}O_{0.98} films. The Hall effect measurements corroborate the trends observed in the electrical properties based on the TeSeO TFTs.

Table S2. Summary of the Hall mobilities and hole concentrations of TeSeO thin films.

Material	Hall mobility (cm²/Vs)	hole concentration (cm⁻³)
Te_{0.9}Se_{0.1}O_{0.98}	60.8	1.1×10^{18}
Te_{0.8}Se_{0.2}O_{0.8}	45.2	8.2×10^{17}
Te_{0.7}Se_{0.3}O_{0.59}	20.5	4.0×10^{17}

Accordingly, the summary of the Hall mobilities and hole concentrations of TeSeO thin films was added to Table S2 on page 9 of the revised Supporting Information. The sentence, "Meanwhile, the Hall effect measurements of TeSeO films show a similar trend to the electrical properties observed in TFTs study (Table S2)." was added to page 5 of the revised manuscript.

10. Fig S3 seem to have Si on Pt side is that due to etching process?

Reply:

We appreciate the referee for pointing out this question. The cross-sectional TeSeO sample presented in original Figure S3 was prepared using a focused ion beam (FIB, FEI Scios Dual Beam system), followed by transmission electron microscopy studies (TEM, JEOL 2100F). Upon thoroughly examining the original EDS mapping data, we found that it is Se (not Si) diffusion to the Pt side, which results from the FIB etching process. The Se diffusion phenomenon during high-energy FIB etching processes is prone to occurrence due to its low melting temperature and high chemical reactivity with Pt.¹⁴

Along with the suggestion raised by another reviewer, we conducted a comprehensive reassessment through EDS mapping measurement and analysis in the revised manuscript, using a spherical-aberration-corrected scanning transmission electron microscopy (STEM) from a cross-sectional view. The STEM high-angle annular dark field (HAADF) image shows good-quality TeSeO film with a thickness of 10 nm. From a microscale, the distributions of Te, Se, and O signals are relatively uniform without noticeable element segregation. The corresponding cross-sectional STEM-HAADF image of TeSeO film was shown in Figure S3 on page 4 of the revised Supplementary Information.

Reviewer #5 (Remarks to the Author):

This is an excellent paper introducing a successful design strategy for p-type Te-Se-O semiconductors fabricated at room temperature that are mechanically and chemically stable. The design basis is clearly explained, the structure of the films is fully examined, and the utility of the material in optoelectronic devices is demonstrated. In the end, the material displays high hole mobility and (for slightly different compositions) a rapid optical response time. I recommend publication in Nature Communications. I do have a few very minor comments/corrections that the authors might address.

Reply:

We appreciate the encouraging comments from the reviewer. The provided feedback is valuable, and we have carefully addressed each comment to improve the quality of the manuscript. Please refer to the point-by-point response below for a detailed overview of the revisions made in response to the comments.

(1) On page 2, the phrase "...have been identified to process respectable..." should probably be "...have been identified to possess respectable..." or some such.

Reply:

We appreciate the referee for pointing out this question. The word "process" has been corrected to "**possess**" on page 2 of the revised manuscript. We have carefully checked the whole article and Supporting Information to avoid similar mistakes.

(2) When discussing the lattice parameters, the HRTEM data is quoted to only 2 digits of accuracy. Within this level of accuracy, the alloy lattice parameters are not slightly smaller

than the Te quoted values, but rather are unchanged. (The comparison with the Se lattice parameters remains correct.)

Reply:

We thank the important comment. We agree that quoting the HRTEM data with two digits of accuracy is more appropriate when discussing lattice parameters. In response to the reviewer's comment, we have modified the discussions about lattice parameters of "The HRTEM image shows clear lattice fringes with lattice spacings of 3.2 Å and 2.2 Å for Te_{0.7}Se_{0.3}O_{0.59}, which corresponds to the (10 $\bar{1}$ 1) and (11 $\bar{2}$ 0) crystalline planes of hexagonal Te/Se, respectively." on page 3 of the revised manuscript, maintaining a two-digit accuracy of lattice parameters in our reporting.

(3) In the penultimate sentence of the paragraph beginning "In general,..." on page 4, the word "monotonously" should be replaced with "monotonically".

Reply:

We appreciate the referee for pointing out this question. The word "monotonously" has been corrected to "monotonically" on page 4 of the revised manuscript.

(4) In figure 5a (and in figure S11a) the key for the principal strain has the phrase (Avg 75%). What does this mean? Also, the peak strain in the model is nearly 4.5%. This is not an insignificant strain - for a typical material, one would see either plastic deformation or fracture before strains this large are observed. Nevertheless, the ligaments in the honeycomb structure appear to withstand these strains experimentally without much change in their optoelectronic response. Do the authors have an explanation for why their material can withstand such high stresses?

Reply:

We appreciate the referee for the valuable comment. In the finite element simulation, the term "the average 75%" serves as an output parameter that indicates the averaging process between the calculation results of neighboring elements. The contributions remain unaveraged if the relative difference between the contributions a specific node receives from neighboring elements exceeds 75%. Conversely, the values undergo averaging if the difference falls below 75%. To simplify this, we have removed this term in the revised manuscript.

Upon comparing the strain distributions of the flat and honeycomb structures, we observed a noticeable reduction of overall strain in honeycomb TeSeO. Specifically, while the flat structure displayed an average strain of approximately 4.5%, the honeycomb structure exhibited an average strain below 3%. More importantly, apart from the region undergoing a tensile state, the rest areas of the honeycomb structure exhibit an even lower local strain. The intensive research over decades has theoretically and experimentally proved that honeycomb structures with nano/micro-scale geometric dimensions can accommodate mechanical deformations and thereby contribute to the superior flexibility of soft (opto-)electronics.¹⁵⁻¹⁷ Because of this, the nanopatterned TeSeO device shows no detectable photocurrent deterioration (Figure 5d) in a bending test with bending times up to 6000. In contrast, the TeSeO flat film without the nanopattern process displayed a significant resistance increase with the bending times (Figure

S12), and eventually, the device broke down because of the appearance of micro-cracks after bending.

To better show the strain distributions of the flat and honeycomb structures, Figure S11 was revised on page 14 of the revised Supporting Information (also shown below). Accordingly, the sentence "benefiting from the porous structure, the strain on the TeSeO honeycomb channel located on the substrate center is efficiently dispersed" was modified on page 6 of the revised manuscript.

Figure S11. (a) FEA simulation of the TeSeO layer/PI substrate model at a bending radius of 1.5 mm. (b) Top view of strain distribution on the bent TeSeO flat film. (c) Top view of strain distribution on the bent TeSeO honeycomb layer.

(5) I am a bit confused by the caption of Fig. S5. In particular, the final sentence discusses a parameter that "decreases" from 0.5 to 3.0. The parameter is Te/Te^{4+} . Should this be Te^0/Te^{4+} and should the value be increasing?

Reply:

We appreciate the referee for the valuable comment. We are sorry for this unclear way of expression about XPS Te 3d peaks in the original manuscript. To avoid the misunderstanding,

the revised manuscript includes a more explicit presentation of the XPS spectra of Te 3d fitting, where Te^0 and Te^{4+} peaks are clearly distinguished at 572.8 eV and 576.0 eV, respectively. The corresponding $\text{Te}^{4+}/(\text{Te}^0+\text{Te}^{4+})$ ratios were determined to be 58%, 42%, 32%, and 25% for $\text{TeO}_{1.16}$, $\text{Te}_{0.7}\text{Se}_{0.3}\text{O}_{0.59}$, $\text{Te}_{0.5}\text{Se}_{0.5}\text{O}_{0.32}$, and $\text{Te}_{0.3}\text{Se}_{0.7}\text{O}_{0.15}$, respectively.

Accordingly, Figure S5 was reorganized, and its caption was modified in the revised Supplementary Information (also shown below). Meanwhile, the sentence "**The corresponding $\text{Te}^{4+}/(\text{Te}^0+\text{Te}^{4+})$ ratios decrease from ~58% ($\text{TeO}_{1.16}$) to ~25% ($\text{Te}_{0.3}\text{Se}_{0.7}\text{O}_{0.15}$) with increasing Se content (Figure S5)**" was revised on page 5 of the revised manuscript.

Figure S5. XPS Te 3d analysis of (a) TeO_{1.16}, (b) Te_{0.7}Se_{0.3}O_{0.59}, (c) Te_{0.5}Se_{0.5}O_{0.32}, and (d) Te_{0.3}Se_{0.7}O_{0.15} thin films. (e) XPS Te 3d_{5/2} peak analysis of TeSeO thin films with different compositions. (f) Extracted Te⁴⁺/(Te⁰+Te⁴⁺) ratios of TeSeO thin films. The coexisted Te⁴⁺ and Te⁰ peaks are clearly distinguished at ~572.8 eV and ~576.0 eV, respectively, which means the partial oxidation of Te. With increasing Se content, the corresponding Te⁴⁺/(Te⁰+Te⁴⁺) ratios decrease from ~58% (TeO_{1.16}) to ~25% (Te_{0.3}Se_{0.7}O_{0.59}), revealing that the Se content can suppress the binding process between Te and O.

List of other changes

- On page 5, the phrase "Table 2 and S1" was revised to "Tables 2 and S3".
- On page 6, the phrase "Figure S12" was revised to "Figure S11".
- On page 8, the phrase "Thermo" was revised to "Thermo Scientific".
- On page 8, the phrase "finite element analysis (FEA)" was revised to "FEA".
- On page 8, the sentence "Elemental mappings were performed using an energy-dispersive X-ray spectroscopy (EDS) detector attached to a spherical-aberration-corrected scanning transmission electron microscopy (STEM, JEOL JEM-ARM300F2)." was revised to the Methods section.
- On the 9, the sentence of "The Ecopia HMS 5300 Hall effect measurement system, equipped with a 0.54 T permanent magnet, was employed to measure the carrier concentration and Hall mobility using the van der Pauw method." was added to the Methods section.
- On page 9, the data availability statement was added: "Relevant data supporting the key findings of this study are available within the article, the Supplementary Information file, and the Source Data file. All raw data generated during the current study are available from the corresponding authors upon request."
- On page 17, the phrase "TeSeO honeycomb layer/PI substrate model" was revised to "TeSeO layer/PI substrate model".
- Since new references (Ref 36 and Ref 37) have been added to the revised manuscript, the sequence of the references was re-arranged.
- Since new supplementary tables (Table S1 and Table S2) have been added to the revised Supplementary Information, the sequence of the supplementary tables was re-arranged.
- In the acknowledgement section, the funding support of "by the General Research Fund (CityU 11204618) and the Theme-based Research (T42-103/16-N) of the Research Grants Council of Hong Kong SAR, China, the National Natural Science Foundation of China (Grant 51672229), the Science Technology and Innovation Committee of Shenzhen Municipality (Grant JCYJ20170818095520778), and a grant from the Shenzhen Research Institute, City University of Hong Kong" is revised to "a fellowship award from the Research Grants Council of the Hong Kong Special Administrative Region, China".

(CityURFS2021-1S04) and the Shenzhen Municipality Science and Technology Innovation Commission (grant no. SGDX2020110309300402; “Modulation and Detection of Terahertz Waves based on Semi-Metallic Two-Dimensional Materials,” CityU”).

References

- 1 Smyth, C. M., Zhou, G., Barton, A. T., Wallace, R. M. & Hinkle, C. L. Controlling the Pd Metal Contact Polarity to Trigonal Tellurium by Atomic Hydrogen-Removal of the Native Tellurium Oxide. *Advanced Materials Interfaces* **8**, 2002050 (2021).
- 2 Kim, T. *et al.* Growth of high-quality semiconducting tellurium films for high-performance p-channel field-effect transistors with wafer-scale uniformity. *npj 2D Materials and Applications* **6**, 4 (2022).
- 3 De Vos, A. & Aerts, J. The influence of deposition rate on the electrical properties of thin tellurium films. *Thin Solid Films* **46**, 223-228 (1977).
- 4 Kim, G. H. *et al.* Room Temperature-Grown Highly Oriented p-Type Nanocrystalline Tellurium Thin-Films Transistors for Large-Scale CMOS Circuits. *Appl. Surf. Sci.*, 157801 (2023).
- 5 Dinno, M. A., Schwartz, M. & Giammara, B. Structural dependence of electrical conductivity of thin tellurium films. *J. Appl. Phys.* **45**, 3328-3331 (1974).
- 6 Kim, T. *et al.* High-Performance Hexagonal Tellurium Thin-Film Transistor Using Tellurium Oxide as a Crystallization Retarder. *IEEE Electron Device Letters* **44**, 269-272 (2022).
- 7 Xu, H. *et al.* High-performance broadband phototransistor based on TeOx/IGTO heterojunctions. *ACS Applied Materials Interfaces* **14**, 3008-3017 (2022).
- 8 Deringer, V. L., Stoffel, R. P. & Dronskowski, R. Thermochemical ranking and dynamic stability of TeO₂ polymorphs from ab initio theory. *Crystal Growth & Design* **14**, 871-878 (2014).
- 9 Raub, S. & Jansen, G. A quantitative measure of bond polarity from the electron localization function and the theory of atoms in molecules. *Theor. Chem. Acc.* **106**, 223-232 (2001).
- 10 Zhang, X., Shao, Z., Zhang, X., He, Y. & Jie, J. Surface Charge Transfer Doping of Low-Dimensional Nanostructures toward High-Performance Nanodevices. *Adv. Mater.* **28**, 10409-10442, doi:10.1002/adma.201601966 (2016).
- 11 Meng, Y. *et al.* Perovskite Core–Shell Nanowire Transistors: Interfacial Transfer Doping and Surface Passivation. *ACS Nano* **14**, 12749-12760 (2020).
- 12 Zhao, C. *et al.* Evaporated tellurium thin films for p-type field-effect transistors and circuits. *Nat. Nanotechnol.* **15**, 53-58, doi:10.1038/s41565-019-0585-9 (2020).
- 13 Bianco, E. *et al.* Large-area ultrathin Te films with substrate-tunable orientation.

- Nanoscale* **12**, 12613-12622 (2020).
- 14 Yu, C.-C. *et al.* Thin-film metallic glass: an effective diffusion barrier for Se-doped AgSbTe₂ thermoelectric modules. *Scientific Reports* **7**, 45177 (2017).
 - 15 Li, Z. *et al.* Flexible transparent electrodes based on gold nanomeshes. *Nanoscale Research Letters* **14**, 132 (2019).
 - 16 Zhu, S., Huang, Y. & Li, T. Extremely compliant and highly stretchable patterned graphene. *Appl. Phys. Lett.* **104**, 173103 (2014).
 - 17 Han, X. *et al.* Nanomeshed Si nanomembranes. *npj Flexible Electronics* **3**, 1-8 (2019).

REVIEWERS' COMMENTS

Reviewer #4 (Remarks to the Author):

The authors have clarified my concerns and queries, the paper can now be accepted.

Reviewer #5 (Remarks to the Author):

I believe that the authors have addressed all my criticisms, as well as those of the other reviewers. I believe the paper is ready for publication in Nature Communications.